# Remodeling of active endothelial enhancers is associated with aberrant gene-regulatory networks in pulmonary arterial hypertension

Armando Reyes-Palomares [1,2], Mingxia Gu [3], Fabian Grubert[4], Ivan Berest[1], Silin Sa[3], Maya Kasowski[4], Christian Arnold [1], Mao Shuai[3], Rohith Srivas[4], Simon Miao[3], Dan Li[3], Michael P. Snyder [4✉], Marlene Rabinovitch[3✉] & Judith B. Zaugg [1✉]

Environmental and epigenetic factors often play an important role in polygenic disorders. However, how such factors affect disease-specific tissues at the molecular level remains to be understood. Here, we address this in pulmonary arterial hypertension (PAH). We obtain pulmonary arterial endothelial cells (PAECs) from lungs of patients and controls (n = 19), and perform chromatin, transcriptomic and interaction profiling. Overall, we observe extensive remodeling at active enhancers in PAH PAECs and identify hundreds of differentially active TFs, yet find very little transcriptomic changes in steady-state. We devise a disease-specific enhancer-gene regulatory network and predict that primed enhancers in PAH PAECs are activated by the differentially active TFs, resulting in an aberrant response to endothelial signals, which could lead to disturbed angiogenesis and endothelial-to-mesenchymal-transition. We validate these predictions for a selection of target genes in PAECs stimulated with TGF-β, VEGF or serotonin. Our study highlights the role of chromatin state and enhancers in disease-relevant cell types of PAH.

[1] European Molecular Biology Laboratory, Structural and Computational Biology Unit, Meyerhofstrasse 1, 69115 Heidelberg, Germany. [2] Department of Biochemistry and Molecular Biology, Complutense University of Madrid, 28040 Madrid, Spain. [3] Department of Medicine, Stanford University School of Medicine, Stanford, CA, USA. [4] Department of Genetics, Stanford University School of Medicine, Stanford, CA, USA. ✉email: mpsnyder@stanford.edu; marlener@stanford.edu; judith.zaugg@embl.de

Complex diseases are often multifactorial with multiple genetic variants associated with an increased risk while monogenic diseases are typically driven by few genetic variants, each of which having a strong effect. An additional complexity—in both complex and monogenic diseases—can come from environmental influences that alter the epigenetic profile in the disease-triggering cells. Pulmonary arterial hypertension (PAH) is an example of a complex multifactorial disease[1], which exists both in a monogenic and polygenic form, and for which the affected cell type is established.

In PAH, the pressure in the lung circulation progressively rises as a result of obliteration of proximal arteries and loss of distal microvessels, and ultimately culminates in right sided heart failure. Patients suffer fatigue and shortness of breath, sometimes for years, but because of the vague nature of these symptoms, the diagnosis of PAH is frequently elusive until the disease is in an advanced stage of pathology. Current treatments, including subcutaneous or intravenous prostacyclin, are vasodilatory in nature; and while they have considerably reduced mortality and improved quality of life[2], the 5-year survival rate of PAH is below 60%[3,4], its diagnosis is increasing in an ageing population with major comorbidities[2] and the underlying causes of the pathology remain unknown. Pursuing these causes will ultimately lead to disease modifying treatments.

Of all patients afflicted with idiopathic (I)PAH, 15% are familial (i.e. multiple family members affected). Of those, 70% carry a mutation in the *BMPR2* gene. Another 20% of sporadic PAH cases also have a *BMPR2* mutation collectively designated hereditary (H)PAH[5]. Mutations in other genes, mostly related to BMPR2 signaling, have been described in a small additional percentage of patients with familial HPAH, including *SMAD9, ACVRL1, ENG, CAV1,* and *KCNK3*[6] and more recently in *ATP13A3, SOX17, GDF2 (BMP9)* and *AQP1*[7]. A large collaborative European cohort of over 1000 patients with IPAH, HPAH and anorexigen associated PAH[7], found that 15% of all patients carried a *BMPR2* mutation, while all other causal variants, were each present in <1.5% of patients[7] at least as judged by whole exome sequencing. While whole genome sequencing studies are underway it will be difficult to relate non-coding variants to disease pathology without a better understanding of the regulatory enhancer landscape, and related epigenetic changes that can drive aberrant gene expression. In addition to the hereditary and idiopathic form of PAH, multiple pre-existing conditions are associated with PAH. Among them are congenital heart disease-associated PAH, the most common subtype (43%) in non-western countries, and schistosomiasis-associated PAH, the third most common subtype (20%) in Brazil[2]. This leads to the hypothesis that in idiopathic and pre-existing condition-associated PAH, epigenetic changes may drive disease progression.

Here, we examine the contribution of epigenetic alterations in primary cells with the aim of deriving insights into the disease mechanism of PAH. We perform RNA-Seq, chromatin mark ChIP-Seq (H3K27ac, H3K4me1, H3K4me3), and chromatin interaction (ChIA-PET) profiling in pulmonary arterial endothelial cells (PAECs) derived from a cohort of IPAH and HPAH patients and healthy controls ($n = 19$). The patient lungs are harvested at the time of transplant, while for the healthy cohort unused donor lungs are used. All PAECs are cultured 3–5 passages before any experiments to minimize procedure-related batch effects. Overall, we find extensive remodeling of the active enhancer landscape (H3K27ac) in PAH patients while the general enhancer mark (H3K4me1), promoters (H3K4me3) and gene expression are not significantly altered, indicating that it is the activity of enhancers that changes. To assess the nature of this epigenetic remodeling we apply diffTF[8] to identify differentially active TFs, which recapitulate almost all TFs that had previously

been implicated in PAH and reveal a large number of previously unreported TFs that are potentially miss-regulated in the disease. To interpret these changes, we devise an approach to build a cell-type specific gene regulatory network (GRN) by integrating H3K27ac, RNA, and ChIA-PET data. This reveals a set of 1880 genes regulated by TFs that are differentially active in PAH (Supplementary Data 5). These genes are enriched for disease-specific phenotypes relevant to PAH and suggest a disease mechanism that primes PAH PAECs to undergo an endothelial to mesenchymal transition (EndMT) in response to endothelial signaling factors. In addition, the GRN suggests changes in gene and protein expression that can be elicited once the PAH PAECs are stimulated with specific growth factors, which we subsequently validate experimentally. Overall, our study illustrates a multi-omics integration approach using active chromatin marks, chromatin conformation, and expression data to gain molecular insights into disease mechanisms. Specifically, we provide a PAH-PAEC-specific gene regulatory network that captures disease biology and suggests a mechanism wherein PAH PAECs are epigenetically primed towards a misguided response to growth factors. This, in turn may impair their angiogenic function, endothelial homeostasis, and lead to EndMT, suggesting that they may take on properties more akin to smooth muscle cells.

## Results

**Extensive remodeling of H3K27ac in PAH patients vs controls.** As the majority of patients with idiopathic PAH harbors no known causal mutation, we wanted to assess the effect of variation in non-coding, potentially gene-regulatory, elements on the disease mechanism. Therefore, we sought to investigate molecular differences in chromatin marks and gene expression in the disease-relevant tissue of PAH patients vs. healthy controls. We performed ChIP-Seq profiling of three active histone marks (H3K27ac, H3K4me1, H3K4me3) and RNA-Seq in small pulmonary artery endothelial cells (PAECs) harvested from individuals suffering from either hereditary PAH (HPAH; carrying a *BMPR2* mutation; $n = 2$) or idiopathic PAH (IPAH; no known mutation; $n = 8$) along with control subjects ($n = 9$; Fig. 1a, Supplementary Fig. 1a, b; Supplementary Data 1). Cells were harvested from lungs and cultured for 3–5 passages to obtain the numbers required for the ChIP-Seq experiments and to minimize any harvesting-procedure related batch effects. Overall, the chromatin marks followed the expected pattern: H3K4me3 and H3K27ac were enriched at transcription start sites (TSS) and correlated with gene expression levels (Pearson's $R = 0.51$ and $R = 0.59$, respectively) while H3K4me1 was more distal to the TSS and less correlated with RNA ($R = 0.22$). This demonstrates that signal between the molecular assays was broadly consistent (Fig. 1b, c). A principal component analysis (PCA) on the most variable regions (top 1000 most variable peaks/genes) for each of the histone marks and RNA revealed that only H3K27ac was able to separate the individuals into PAH patients and healthy controls (Fig. 1d; Supplementary Fig. 1c). Consistently, we found thousands of loci that are significantly differentially modified for H3K27ac (11,701 at 5% false discovery rate (FDR), Supplementary Data 2) between IPAH and controls (Fig. 1e and Supplementary Fig. 1e), while very few regions were differentially modified for H3K4me1 and H3K4me3, and no differentially expressed genes were found at FDR 5% and only a handful was significant at 10%. We observed similar fold-changes between HPAH patients against controls (Supplementary Fig. 1f) and thus combined the IPAH and HPAH patients for all subsequent analyses. Since previous studies had identified a handful of differentially expressed genes on different sample sets[9–11], we performed validation qPCR in a subset of the individuals (two IPAH,

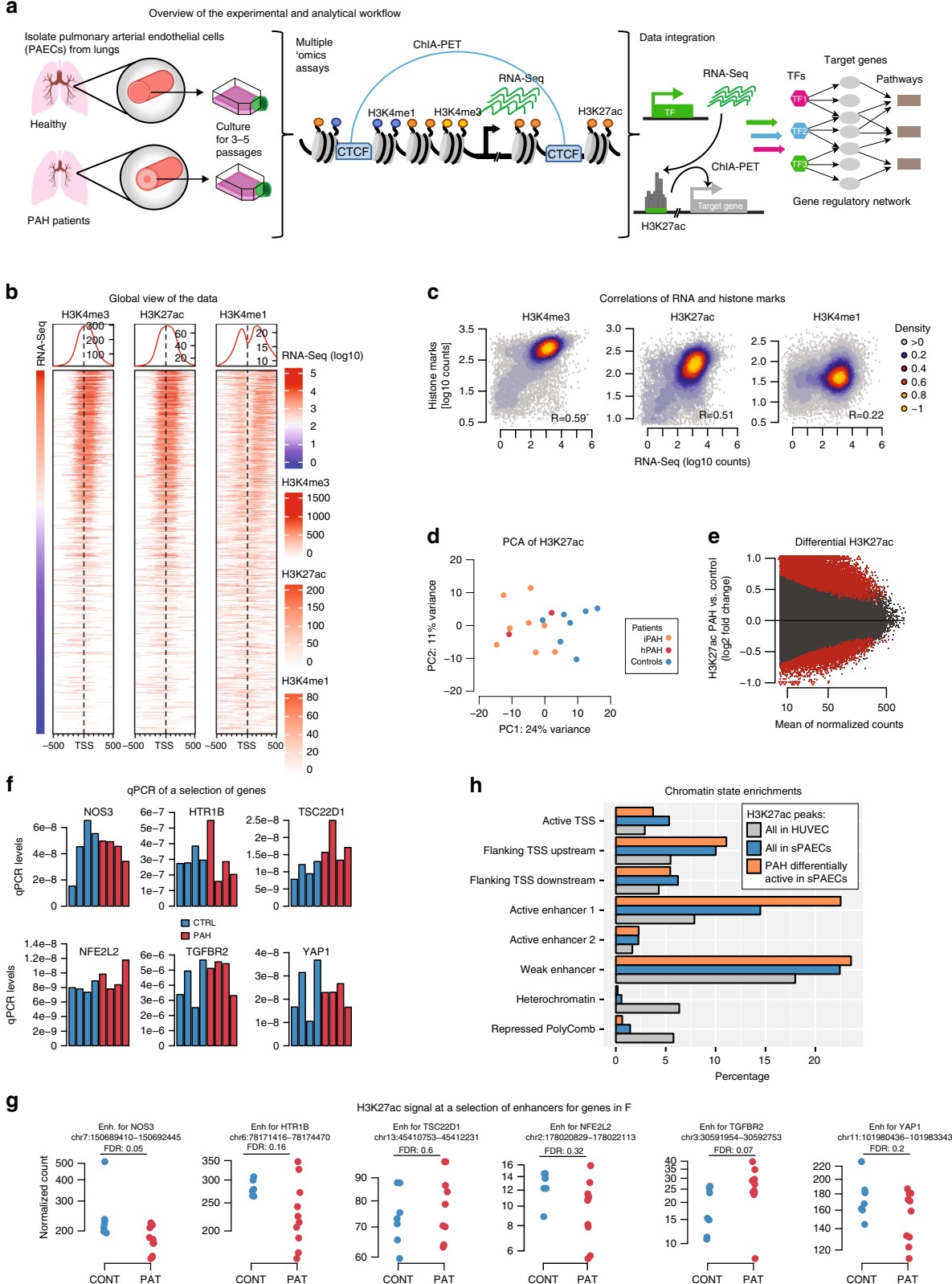

two HPAH, four controls) on the baseline expression levels for some genes (Fig. 1f) that were selected to have a range of differential H3K27ac signals in their promoters (Fig. 1g). None of them showed any difference in baseline gene expression levels in the qPCR assays. When we compared pair-wise correlations between individuals for histone marks and RNA we found a significantly higher variation for RNA pairs, regardless of whether the individuals came from PAH or controls, indicating that the lack of differentially expressed genes may partially be explained by sample heterogeneity (Supplementary Fig. 1g).

To gain functional insights into the differentially modified H3K27ac regions we assessed their chromatin states using data

**Fig. 1 Histone modification and gene expression changes in PAH. a** Overview of the experimental and analytical workflow of the study. **b** ChIP-Seq signal distributions for H3K4me3, H3K27ac, and H3K4me1 around the TSS of expressed genes in PAECs are shown as heatmap. The normalized read counts (mean across all individuals) for each histone are colored according to their specific legend. TSS regions are sorted by the average expression level of the gene. **c** Pairwise correlations between gene expression and H3K27ac, H3K4me3, H3K4me1 ChIP-Seq signal of peaks within 2.5 kb of genes are shown. The strength of the association was estimated from Pearson's correlation coefficient ($R = 0.51$, $R = 0.59$, $R = 0.22$, respectively). **d** Principal component analysis (PCA) of H3K27ac ChIP-Seq signal of the 1000 most variable regions across all individuals is shown. Individuals are colored by disease status: idiopathic PAH (orange), hereditary PAH (red) and control (blue). **e** Log2-ratio vs mean expression (MA plot) is shown for the differential analysis of H3K27ac signal between patients and controls; red dots represent significantly modified regions (FDR < 5%; $n = 10$ PAH and seven controls). **f** RT-qPCR signals for baseline experiments for a selection of genes and individuals are shown ($n = 4$ controls and 4 PAH). Source data are provided in Source Data File. **g** H3K27ac signal at enhancers (Enh) of the genes assayed by qPCR in **f** are shown per individual. **h** Distribution of chromatin states of H3K27ac regions detected in PAECs (blue bars, representing 263,910 genomic features) and in differentially modified regions (orange bars, representing 31,084 genomic features). Chromatin states were obtained from human umbilical vein endothelial cell (HUVEC; grey bars, representing 515,807 genomic features) from the Roadmap Project[12] (file: E122_18_core_K27ac_dense.bed).

from the Epigenomic roadmap project[12]. We used the chromHMM states for human umbilical vein endothelial cells (HUVEC), which is the cell line closest resembling PAECs. As expected for active marks, we found that the H3K27ac peaks fell mainly in active regions (including promoters and enhancers) and were absent from heterochromatin (Fig. 1h, Supplementary Fig. 1h). In contrast, the differentially acetylated regions were only enriched in active enhancers (adjusted $p$-value (adj. $p$-val) $< 1E - 16$; Odds Ratio (OR) $= 4.4$) and even slightly depleted from active promoters (adj. $p$-val $= 0.03$; OR $= 0.93$; Fig. 1g).

Together, these results suggest that while PAECs from PAH patients and healthy controls are similar in terms of gene expression, they have been undergoing a large-scale remodeling of the active chromatin regions, in particular at enhancers. This could indicate that while still being in an endothelial state, the PAH PAECs might be primed to respond differently from the healthy cells to external stimuli.

**Differentially H3K27-acetylated regions are disease relevant.** To delineate the relationship between the differentially modified regions and molecular pathways relevant to PAH, we performed a functional enrichment analysis of the genes that are potentially regulated by the differentially H3K27-acetylated regions. Yet, consistent with the depletion of differentially modified regions in promoters, we found no enrichment for specific GO terms and only a few very broad categories enriched among the genes that had a differentially H3K27-acetylated peak in their promoter (Supplementary Fig. 2a). Therefore, and since the differential H3K27ac regions were enriched in enhancers, we had to devise a strategy for linking the H3K27ac regulatory elements to their target genes.

In a first step we performed chromatin interaction analysis by paired-end Tag sequencing (ChIA-PET) with CTCF as anchor factor in a pool of PAECs from healthy donors (see Methods). The data quality was above ENCODE standards and in a similar range to ENCODE ChIA-PET (Supplementary Fig. 2b). This resulted in 21,805 chromatin loops with a mean size of 283.7 kb (Fig. 2a), each containing on average two genes and 25 H3K27ac peaks (Supplementary Fig. 2c). We observed that the H3K27ac peaks within a ChIA-PET loop behaved in a highly coordinated manner, either being all up- or all down-regulated between patients and controls (Fig. 2b). This coordination of PAH-associated chromatin changes across entire CTCF loops suggested that they may act as independent regulatory units. Thus, instead of requiring a direct enhancer-promoter contact, we sought that we can use the highly coordinated loops, that we termed chromatin regulatory domains (CRD), as a basis for searching for enhancer-target gene interactions. To identify the CRD, we used the correlation of H3K27ac signal at both boundaries of a ChIA-PET loop as proxy for the loop's coordination strength,

assuming that if the most distally located H3K27ac peaks of a loop are correlated, all peaks in-between will be correlated as well. A loop was then defined as CRD when the boundary H3K27ac peaks were positively correlated with a nominal $p$-value < 0.05 (Fig. 2c). A similar concept of CRDs has recently been proposed in lymphoblastoid cell lines[13]. As a statistical validation we showed that the $p$-value distribution of negative correlations showed no signal while positive correlations were enriched for low $p$-values (Fig. 2d). Then, to link genes to their specific regulatory elements (enhancers, promoters), we used Pearson correlations between the H3K27ac signal and RNA expression of each gene-peak pair within each CRD, and defined positive correlations (at a nominal $p$-value of 0.05) as regulatory, enhancer-gene and promoter-gene, interactions, assuming that enhancers and gene expression should be positively correlated. Again, we found more signal for positive correlations, while the $p$-value distribution for negative correlations and randomized gene-peak links showed no signal (Fig. 2e). The number of enhancers per gene, defined as H3K27ac peaks that correlate with gene expression, was higher in CRDs than non-CRD ChIA-PET loops, thus validating our assumption (Fig. 2f, Supplementary Fig. 2c, d). Examples of genes within CRDs are shown in Fig. 2g and Supplementary Fig. 2e.

To assess the biological significance of CRDs we performed a GO enrichment analysis on genes linked to differentially active elements within CRDs and compared it to a GO analysis of genes linked within non-CRD ChIA-PET loops. For the CRD-linked genes we found strong enrichments for endothelial biology and processes related to PAH, such as "endothelial cell proliferation" (adj. $p$-value $= 0.03$, OR $= 4.5$) and "endothelial cell migration" (adj. $p$-value $= 0.03$, OR $= 3.8$; Fig. 2h), while no GO term was significantly enriched for non-CRD linked genes. This highlights the biological relevance of CRDs for investigating enhancer–gene regulatory interactions.

Overall, these results show that linking genes to differentially active enhancers in PAH via CRDs captures the expected disease-relevant and cell-type specific signals and illustrates the importance of linking enhancers to their target genes for understanding disease mechanisms. At the same time, finding these endothelial and PAH-specific functions, demonstrate the suitability of our data and analysis approach to learn more about the pathobiology involved in PAH.

**Known PAH TFs are differentially active in PAH vs controls.** Given the strong enrichment in differentially active regions within CRDs for endothelial function and disease-specific GO terms we wanted to understand more about the molecular pathways leading to these enrichments. Based on findings of a previous study, in which we found that genetic variants that control histone mod-ification levels often lie within TF binding sites[14], we hypothesize

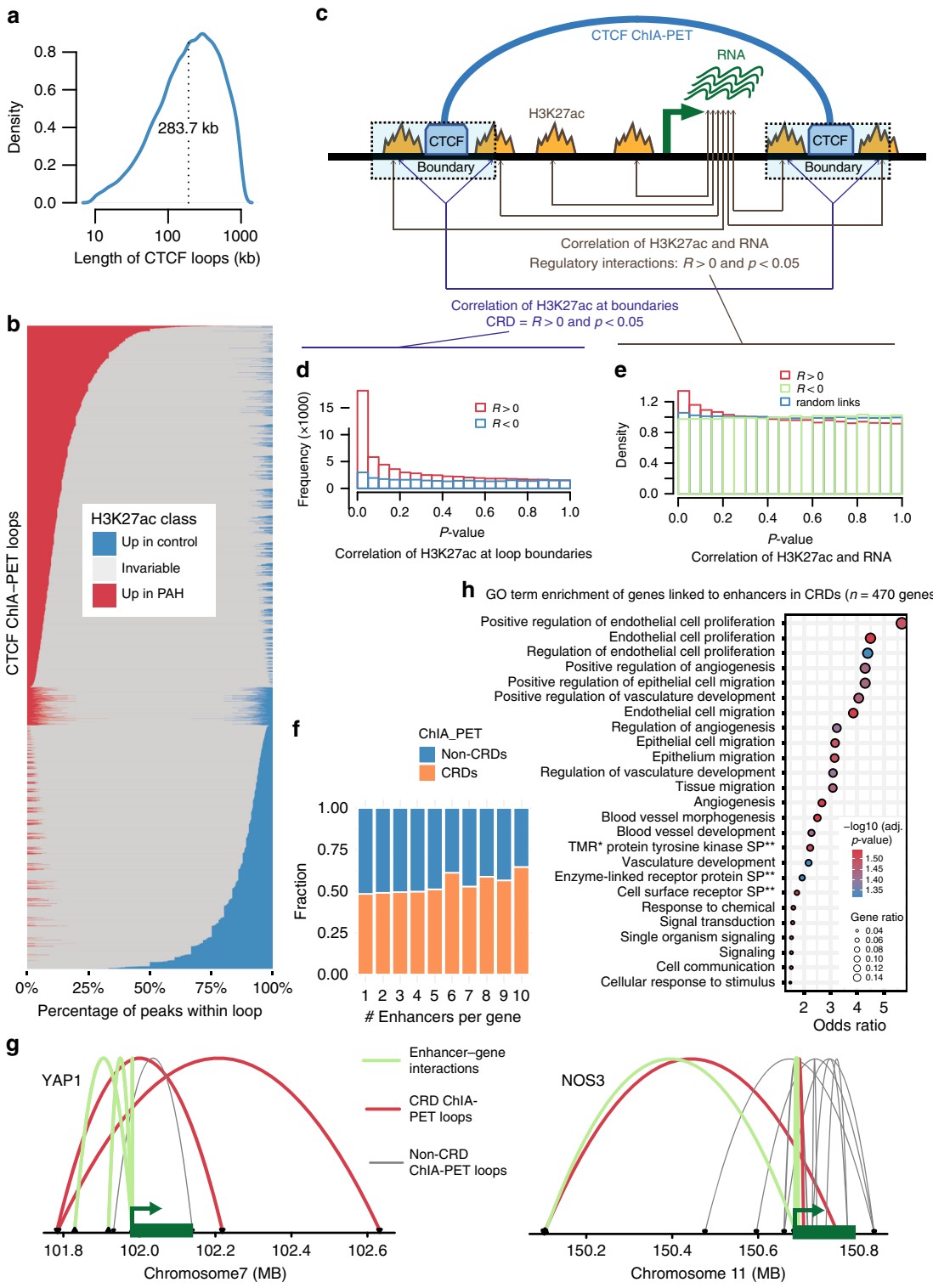

**h** GO term enrichment of genes linked to enhancers in CRDs (*n* = 470 genes)

that differences in H3K27ac signal are driven by a specific set of disease-associated TFs. We have recently formalized this idea in a tool called diffTF to estimate differential TF activity based on genome-wide H3K27ac signal between two groups of samples[8]. Briefly, diffTF aggregates differences in H3K27ac signal in proximity of all putative binding sites of a given TF as estimate of its differential activity (see Methods, Supplementary Fig. 3a, b[8]). Here, we applied diffTF to quantify differential activity between PAH patients and controls for all TFs that are expressed in PAECs in our data. In total, we found 318 differentially active TFs (represented by

339 out of 640 tested motifs; FDR 0.001; Fig. 3a, Supplementary Data 3). Reassuringly, when we classified the TFs into activators and repressors (using diffTF in classification mode[8]) we found an overall significant correlation between differential TF activity and differential expression between patients and controls (Supplementary Fig. 3c), despite the lack of significantly differentially expressed genes as reported above. This indicates that small changes in expression of a TF can lead to detectable differences in activity probably due to the aggregating H3K27ac signal across all binding sites[8].

**Fig. 2 Chromatin regulatory domains (CRDs) and gene-enhancer interactions. a** Distribution of lengths is shown for CTCF-mediated chromatin loops (ChIA-PET; $n = 21,805$). **b** The fraction of differentially H3K27ac modified peaks are shown per CRD. Peaks were classified as higher in PAH (red), healthy (blue), or unchanged (grey) if their log2 fold change value was above 0.5, below −0.5, or between −0.5 and 0.5 respectively. **c** Schematic of the approach for determining CRDs: H3K27ac peaks at the two boundaries of CTCF-mediated ChIA-PET loops are positively correlated ($p$-value < 0.05). **d** $P$-value distribution of Pearson correlation coefficients (R) between H3K27ac signals located at opposite anchor points within the same CTCF loop; $R > 0$ (red bars) and $R < 0$ (blue bars). **e** $P$-value distribution of Pearson correlation coefficients (R) of H3K27ac signal and expression of genes that are within the same CTCF loop; $R > 0$ (red bars), $R < 0$ (green bars) and randomly located H3K27ac signals (blue bars). **f** GO terms that are enriched among the genes that are connected to differentially H3K27-acetylated enhancers within CRDs. Some of the GO terms have very similar sets of genes and are thus not completely independent. **g** For each number of enhancers (x-axis) the fraction of genes within CRDs (orange) and within non-CRD ChIA-PET loops (blue) are shown. **h** Examples of ChIA-PET data is shown for two genes also shown in Fig. 1f (additional genes shown in Supplementary Fig. 3e). CRD-ChIA PET loops are shown in red, correlation-based enhancer-gene interactions are shown in green and non-CRD ChIA-PET loops are shown in grey. Source data for **a**–**b**, **d**–**e**, **h** are provided in Source Data File.

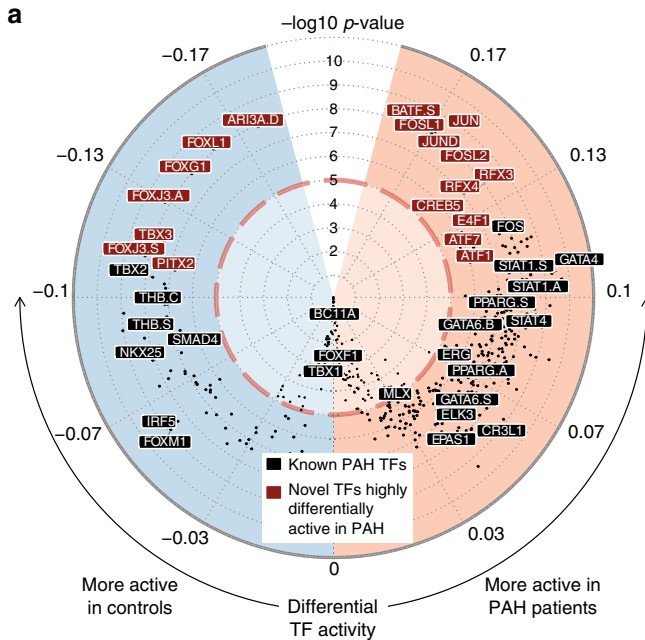

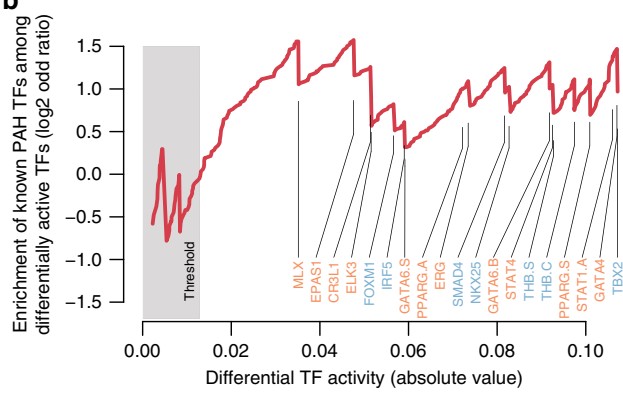

**Fig. 3 Differential TF binding activity between PAH and controls. a** Circular vulcano-plot shows the distribution of differential TF activity based on H3K27ac signal differences between PAH and controls. TFs previously associated with PAH are labeled in black. TFs that are more differentially active between patients and controls than the known PAH-TFs are labeled in red. Full list of differentially active TFs are provided in Supplementary Data 3. **b** Enrichment of known TFs related to PAH among the set of differentially active TFs for increasing thresholds for defining differential TF activity. TF activity is shown on an absolute scale with TFs being more active in patients and controls colored in orange and blue, respectively.

Overall, out of the 20 TFs that have previously been associated with PAH (compiled from https://monarchinitiative.org, https://www.opentargets.org/, and ref. [15]; see Methods; Supplementary Data 4), 16 were identified as differentially active, and they were strongly enriched among the differentially active TFs regardless of the threshold used to define differentially active TFs (Fig. 3b). This shows that our method to assess TF activity is both robust and sensitive to identify TFs that are potentially important drivers of the disease. SOX17, which was only recently identified as PAH TF[7,16] was also significantly miss-regulated in our PAH patients (Supplementary Data 4).

A number of TFs (17 represented by 18 motifs; Fig. 3a) showed an even stronger differential activity between PAH and controls than the known PAH-TFs, which suggests that they could be novel candidates contributing to the disease. These newly identified TFs include the AP1 complex (BATF, FOSL1, JUN, JUND, FOSL2 as well as ATF1 and ATF7, all of which have similar binding sites and should thus be considered indistinguishable), members of the RFX family (RFX3, RFX4, again similar motifs), CREB5, and E4F1 with higher activity in patients, and ARID3A, members of the FOX family (FOXL1, FOXG1, FOXJ3, again similar motifs), TBX3 and PITX2 with higher activity in the healthy individuals.

The AP1 complex, a known pro-inflammatory factor[17], is particularly interesting in the light of our earlier findings that inflammation plays an important role in PAH[18,19]. Similarly, E4F1 and ARID3A have been shown to interact with and activate specific functions of p53[20,21], which we have recently shown to be upregulated upon loss of BMPR2 in a process that is linked to the inflammasome[22]. CREB5 has been proposed as a hypoxia response gene in lung[23], and its enhanced activity is likely a consequence of the disease. TFs of the RFX family have long been linked to repression of collagen[24], which in turn is a known PAH associated gene (Supplementary Data 4). Finally, mutations in TBX4, which has a highly similar binding site to TBX3 has been associated with childhood-onset PAH[25]. These examples illustrate that many of the novel predicted PAH-TFs may indeed play a role in the pathogenesis and provide important starting points for further investigations.

**Gene regulatory network and PAH-specific TFs highlight PAH biology.** We next wanted to investigate the genes that are regulated by the differentially active TFs in PAH. For this, we attempted to generate a PAH-specific gene regulatory network linking TFs to their target genes. Above, we have already defined the links between genes and their enhancers (Fig. 2c), so we only needed to link TFs to the regulatory elements they modulate. To do so, we devised a correlation-based approach, which links TFs to their target regulatory element if (i) the regulatory element has

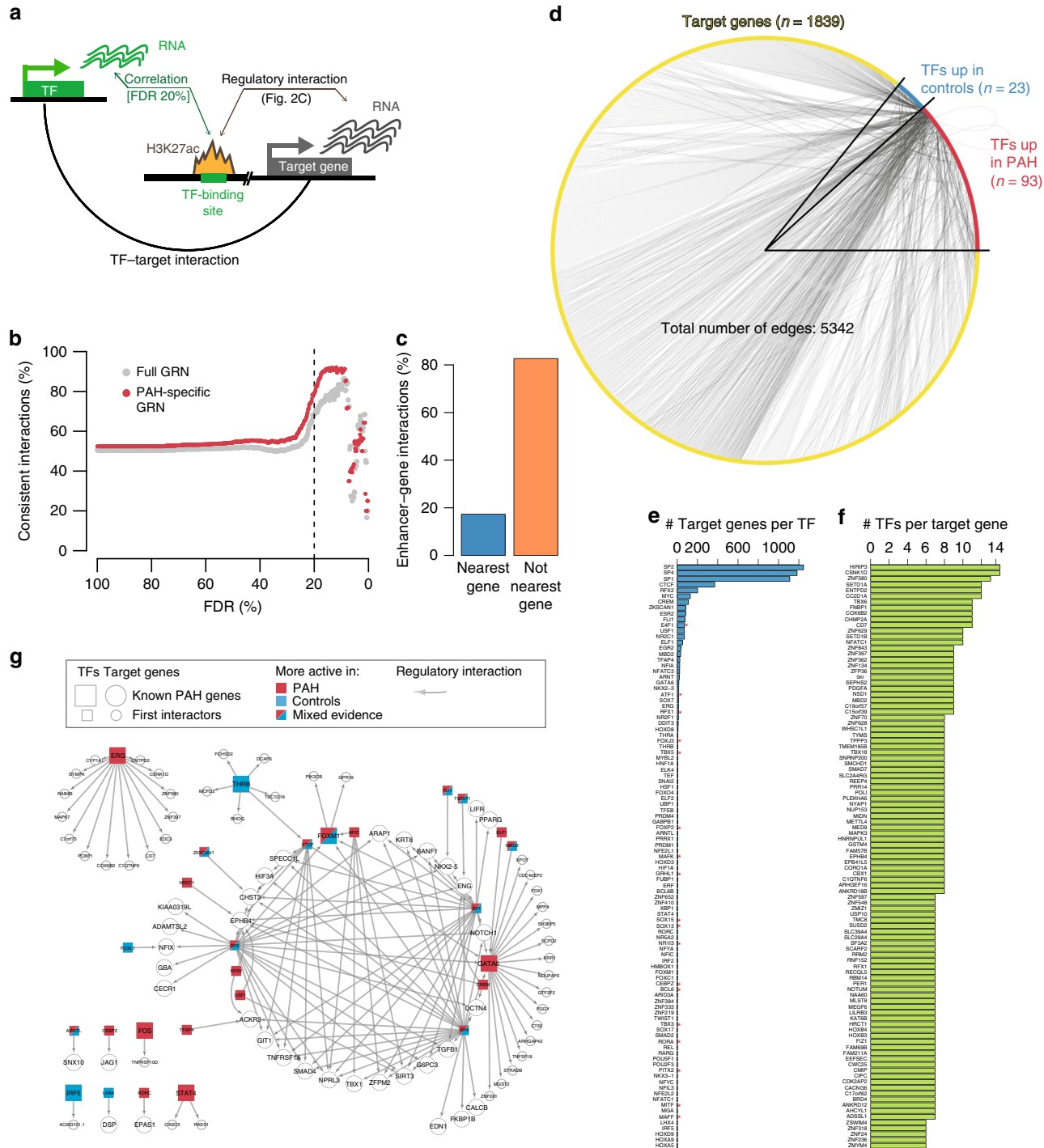

**Fig. 4 Gene Regulatory Network for PAH based on TF-enhancer-gene links. a** Schematic representation of the gene regulatory network (GRN) approach. **b** Fraction of consistent interactions defined as regulatory interactions where the TF activity varies in the same direction as the H3K27ac signal at the putative target peak. The consistency increased considerably around the 20% FDR threshold (see Methods). **c** Percentage of regulatory interactions mediated by enhancers that are connected to the nearest transcription start site (TSS; blue, $n = 1289$) vs other TSS (orange, $n = 6153$). **d** GRN in PAH. Nodes represent TFs (square) and target genes (circle) that are within chromatin regulatory domains (CRDs). TFs more active in PAH (red), TFs more active in controls (blue), target genes more active in PAH (orange) and target genes more active in control (green). The color of the edges represents a positive (green) and negative (red) correlation for TF expression and H3K27ac signal at the promoter of the target gene. TFs with more than 100 connections were removed to improve clarity of the network visualization. The network is provided in Supplementary Data 5. **e** Ranked TFs by their number of interactions in the PAH genetic regulatory network. **f** Ranked target genes by their number of interactions in the PAH genetic regulatory network, to make clear the plot only the top 100 target genes are represented. **g** Detailed sub-network of the PAH-GRN of known genes associated with PAH (large nodes) and their first interactors (small nodes).

a putative binding site of the TF and (ii) was significantly (positively or negatively, reflecting activators and repressors) correlated with the expression level of the TF (Fig. 4a; Methods). This assumes that any change in expression of a TF should result in a change in activity of the enhancers and promoters it is regulating. Note that this approach is based on the co-variation of TF expression and enhancer signal across individuals and independent of the differential expression/signal between conditions. To estimate the FDR of the TF-peak links we used the correlations of TFs with peaks that do not contain their motifs as background to calculate an empirical FDR (Methods.) We chose an FDR of 20% for the network since this seemed to maximize the fraction of "consistent" interactions whereby consistent was defined as interaction where the activity of a TF and its target peak go in the same direction (Fig. 4b). Notably, we observed that <20% of the enhancers are linked to the closest gene (in terms of distance to TSS), which again highlights the importance of a cell-type specific gene–enhancer linking (Fig. 4c).

To focus on disease-specific interactions, we selected the sub-network that is regulated by the putative PAH-TFs we obtained above. This resulted in a PAH-specific gene regulatory network comprising 116 TFs, 1,880 target genes and 5,342 interactions (Fig. 4d–f, Supplementary Data 5). To assign directionality to the network, we used the TF activities calculated by diffTF based on motifs and, if available, ChIP-Seq data (see Methods). TFs, for which the directionality was inconsistent between motifs and ChIP-Seq data, but were significant in at least one of the analyses were labeled as "mixed evidence". This network was then used to query the regulatory relationships between specific sets of genes.

As a first example, we visualized the regulatory interactions that involve the previously known PAH-associated TFs (Fig. 4g, Supplementary Data 4). Of the 306 known PAH-associated genes 209 were expressed in our data, and of these our GRN covers 9 TFs and 45 target genes. Overall, we found that the genes group into one main regulatory cluster dominated by GATA6 and SP1, SP2 and SP4, and several smaller clusters regulated by STAT4, ERG, and THRB among others. This indicates that while there are separate regulatory pathways affected, many known PAH genes are involved in the same regulatory cascade.

To assess the biological significance of the gene regulatory network we performed GO enrichment analysis on the target genes, using all expressed genes as a background, and excluding TFs. Overall, we found over 250 GO terms significantly enriched among the target genes (Supplementary Data 6; subset displayed in Fig. 5a). Many of the enriched biological processes are associated with endothelial biology and have been implicated in PAH. Examples include cell migration and tube morphogenesis ("regulation of cell migration" (OR = 1.4, adj. $p$-val = 4.1e-3) and "regulation of tube size" (OR = 2.5, adj. $p$-val = 0.011; Supplementary Data 6)), which are impaired in PAH-derived endothelial cells. Other terms, such as "cellular response to corticosteroid stimulus" (OR = 3.2, adj. $p$-val = 0.02) and "regulation of vasoconstriction" (OR = 3.7, adj. $p$-val = 0.016) might be a consequence of the disease and the treatment the patients obtained prior to their lung transplant since all patients were treated with some kind of vasodilator drug (Fig. 5a, Supplementary Data 1). Among the most specific GO terms we identified was "SMAD protein complex assembly", a process affected by the *BMPR2* mutation that causes HPAH (Fig. 5b).

Overall, these terms demonstrate that our regulatory network, driven by differentially active TFs in PAH patients, is able to recover the expected biological processes, thus validating the approach.

**Gene regulatory network suggests disease mechanism in PAH.** Next we used the PAH regulatory network to uncover processes that integrate disease mechanisms. The most enriched GO term among the PAH-specific regulatory network was "positive regulation of smooth muscle cell differentiation", for which we identified 7 out of 10 genes (OR = 17, adj. $p$-val = 1.9e-3; Fig. 5a). We also found other muscle biology related processes along with "epithelial to mesenchymal transition", and "mesoderm development" among the most enriched terms (Fig. 5c). Querying the GRN for the directionality of activity for the TFs in these GO terms revealed that the majority of TFs regulating them are more active in PAH, thus indicating that these pathways are upregulated in PAH. Together with the enrichment of "cell proliferation" (OR = 1.4, adj. $p$-val = 4.9e-3) this may suggest a mechanism where proliferating endothelial cells start differentiating into smooth muscle-like cells. One clue of how this could happen comes from the term "response to transforming growth factor beta" (OR = 1.8, adj. $p$-val = 0.045). It suggests an aberrant response of the endothelial cells to growth factor stimulation, which may lead to cell proliferation and EndMT, and may ultimately promote the over-proliferation of smooth muscle cells typically observed in PAH[26]. Our regulatory network indicates that TGFβ1, also comprised in the smooth muscle differentiation GO term, is controlled by the TFs of the SP family (SP1, SP2, SP4) that are differentially active in PAH patients (Fig. 5d). More importantly, we found that the SMAD2 TF, which is activated through TGFβ, is strongly active in PAH (Fig. 5e), thus suggesting a miss-regulation of the TGFβ pathway in the endothelial cells of PAH patients. It also suggests a previously unknown link between TFs that are related to inflammation (SP family), signaling (SMADs), and EndMT (SNAI1).

Altogether our data indicate that the active chromatin landscape in PAH patients is specifically altered at enhancers that regulate genes involved in smooth muscle cell differentiation and may respond aberrantly to TGFβ signaling. This would also explain the lack of relationship between the RNA and the H3K27ac mark described in the beginning. We would thus expect that, upon stimulation of endothelial growth or signaling factors, some of these target genes would dramatically differ in their expression response in PAH patients and controls.

**RNA response to growth factor reflects steady state H3K27ac.** To test the hypothesis that enhancers in PAECs of PAH patients are priming specific genes to respond aberrantly to growth factor stimulation, we measured the response to three stimuli relevant in PAECs in a set of genes that were predicted to be primed differently in PAH patients. The genes were selected to represent different processes that we found enriched in the PAH GRN, and were required to be linked (in our GRN) to a differentially H3K27-acetylated peak. Specifically, we chose NOS3 and HTR1B (vascular system), TSC22D1 (basic transcriptional processes), YAP1 and DKK1 (mesoderm development), and NFE2L2 (response to stimuli) and TGFβR2 (direct target of TGFβ). We used eight PAECs lines—four from patients (two IPAH and two HPAH) and four from controls—and subjected them to stimulation with three different growth factors/signalling molecules that are relevant for endothelial cells: vascular endothelial growth factor (VEGF; 10 ng/ml), transforming growth factor beta (TGFβ; 10 ng/ml), and serotonin (10uM), each for 24 h. We then performed qPCR on the selected genes and compared the expression of PAH vs controls.

As expected from the global RNA-Seq data no significant changes were detected between patients and controls in unstimulated PAECs (Fig. 6a). Instead, however, we found that most genes showed dramatic differences between patients and controls in at least one of the stimulations (Supplementary Fig. 4a). Importantly, we found a strong correlation between

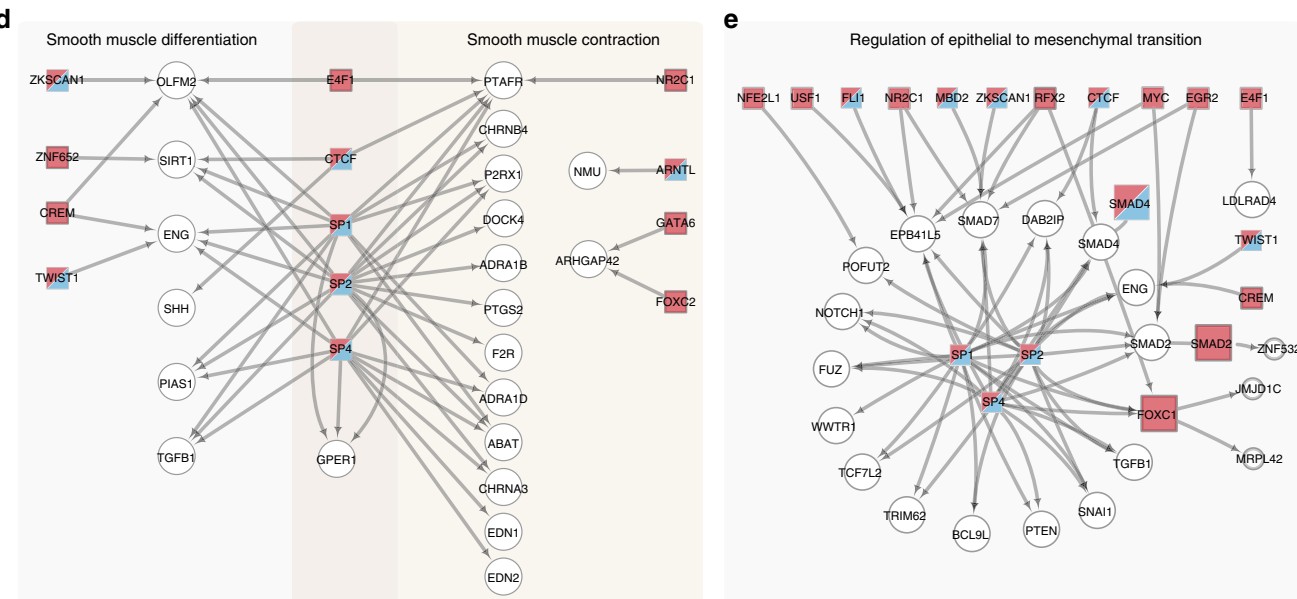

**Fig. 5 Functional analysis of genes targeted by differential PAH-TFs. a** Enrichment of gene ontology (GO) annotations for target genes in PAH-specific GRN (adj. *p*-val < 0.05). Only GO terms that have between 3 and 80 genes are shown and some highly redundant terms were removed (full list in Supplementary Data 6). All expressed genes were used as background. **b** Network of target genes associated with the GO term "SMAD complex assembly" (large nodes) and their first interactors (small nodes). **c** GO terms related to "muscle" and "mesenchyme" that were enriched among the target genes of the PAH-specific GRN. **d** Network of target genes associated with the GO terms "Smooth muscle differentiation" and/or "Smooth muscle contraction" (large nodes) and their first interactors (small nodes). Color code as in **b**. **e** Network of target genes associated with the GO terms "Regulation of epithelial to mesenchymal transition" (large nodes) and their first interactors (small nodes). Color code of the nodes as in **b**.

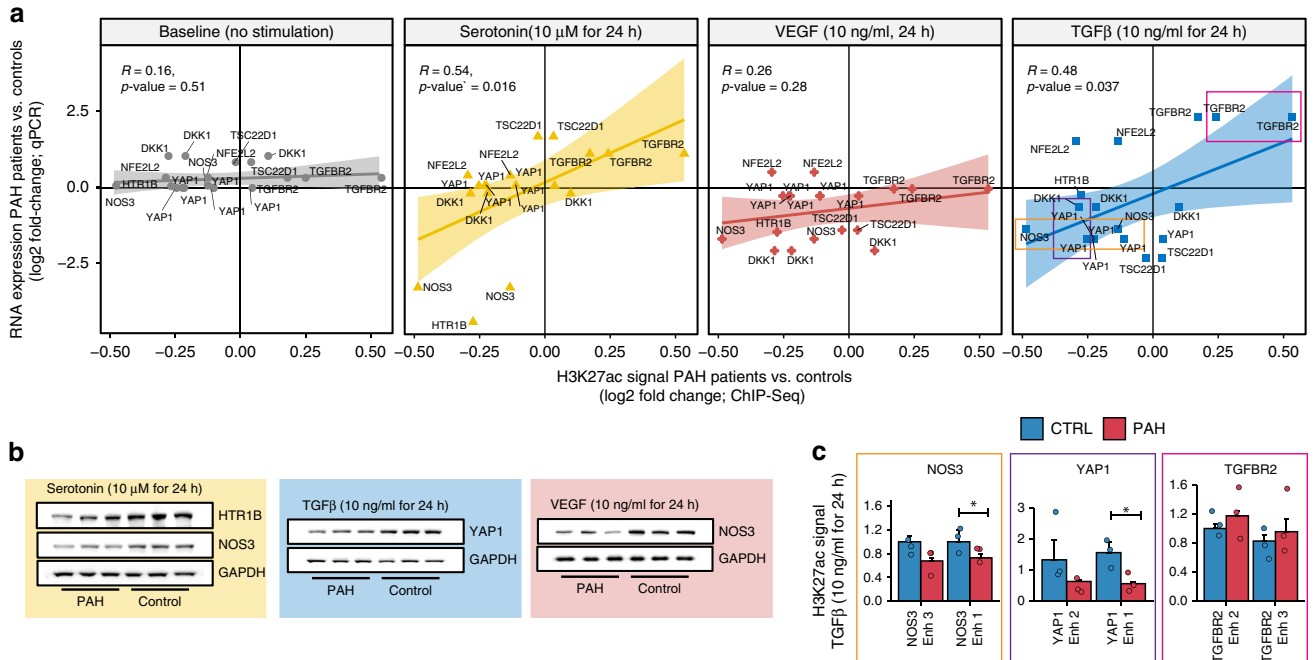

**Fig. 6 Experimental growth factor stimulations in PAECs confirms priming. a** Log2 fold-change of the gene expression (as measured by RT-qPCR) and the H3K27ac signal (as measured by ChIP-Seq) of the regulatory element(s) linked to the gene are shown as scatter plot. Each point is a regulatory element-gene pair. The fold-changes are calculated between PAH patients and controls in steady state for H3K27ac and after a stimulation with endothelial-specific growth factors (serotonin—yellow triangles, vascular endothelial growth factor (VEGF)—red crosses, and transforming growth factor β (TGFβ)—blue squares) for the RNA. Pearson correlation coefficients (R) and associated p-values are given in the plots. (qPCR experiments: n = 4 patients, four controls). **b** Western blot analyses are shown for a selection of genes that showed a differential response to any of the stimulations on the RNA level. GAPDH levels are shown as a reference. Source data are provided in Source Data File. **c** ChIP-qPCR analysis for enhancers predicted to prime response in PAH patients. H3K27ac signal at selected enhancers (Enh; orange, brown and pink box for NOS3, YAP1, and TGFBR2), was measured after TGFβ stimulation in patient and controls. All of the enhancers showed a trend in the predicted direction and some of them statistically significant (n = 3 patients, three controls). Error bars indicate standard deviation across three replicates.

changes in enhancer activity (H3K27ac) at baseline and changes in expression of their target genes under endothelial signaling when comparing PAH and controls (Fig. 6a). This corroborates the notion that these genes are being primed by enhancers to respond in an aberrant way in PAH patients, which could then lead to abnormal growth or differentiation. For the TGF-beta stimulation we found that the enhancers of the differentially regulated genes become even more differentially active upon stimulation (Fig. 6b). To extend the functional importance of these observations we also showed that protein levels (measured by western immunoblot) for some of the genes in each stimulation concurred with the mRNA levels (Fig. 6c). For YAP1, which is highly regulated by phosphorylation, we also verified that the phosphorylated form changed in accordance with its RNA level (Supplementary Fig. 4b).

The following examples serve as a proof of concept for interpreting the data in the context of PAH disease mechanism: First, VEGF normally induces NOS3[27]. In PAH PAECs its enhancer has reduced H3K27ac signal relative to control PAECs. Consistently, the NOS3 mRNA and protein response to VEGF is increased only in control PAECs. Second, serotonin normally stimulates angiogenesis through the HTR1B receptor C (*HTR1BRC*). In PAH PAECs, however, an enhancer of HTR1BRC shows reduced H3K27ac signal, and seems to prime the cells against the induction of *HTR1BRC* mRNA and protein after serotonin stimulation, thus offering another way in which angiogenesis is subverted in PAH, that is consistent with the reduction in NOS3 in response to serotonin. Alternatively, the reduction in H3K27ac marks on *NOS3* enhancers may alter its response in PAH. Third, YAP1 promotes angiogenesis in

endothelial cells[28] but *YAP1* is not known to be regulated by TGFβ in endothelial cells[29]. Here too the impaired upregulation of YAP1 as well as NOS3 in response to TGFβ in PAH PAEC might underlie the impaired angiogenesis in these cells. Finally, novel TFs, such as NFE2L2 and TSC22D1, appear aberrantly upregulated by TGFβ and may relate to new mechanisms perturbing this pathway. Thus, the H3K27ac marks help us uncover new pathways and effectors important in endothelial regeneration in response to stimulation that are compromised in diseases, such as PAH.

Overall, these experiments validate our predictions that endothelial cells in PAH patients are primed to respond in a significantly different way to growth factor stimuli than in healthy controls in a mechanism that likely involves enhancer-priming. Taken together, our data provide the epigenetic basis for aberrant AP1, TGFβ, serotonin and VEGF signaling that conspire to cause inflammation, aberrant angiogenesis and EndMT, and the propensity for smooth muscle cell proliferation, all features of PAH. Moreover, our results suggest that potential therapies should be considered not only by their impact on cell surface receptor or signaling pathways, but on whether they reverse the sequelae of the aberrant epigenetic landscape.

## Discussion

Our study shows that small pulmonary arterial endothelial cells (PAECs) from PAH patients display remodeling of their enhancer landscape while not showing any significantly differentially expressed genes at steady state. This apparent contradiction led us to propose that enhancers may prime the PAECs in PAH to respond aberrantly to endothelial signaling, which we confirmed

experimentally for a number of genes. Linking genes to their proximal and distal regulatory elements using chromatin conformation and a correlation-based approach, we found specific endothelial and PAH-related processes that indicate that PAECs in PAH are primed towards an endothelial to mesenchymal transition in response to normal endothelial signaling. This highlights the importance of linking genes to their distal regulatory elements to uncover biological functions.

Converting global H3K27ac signal into TF activities using diffTF[8], we identified a large number of differentially active TFs between patients and controls, including almost all previously known PAH-associated TFs.

Integrating RNA, ChIA-PET and H3K27ac data we have devised an approach to generate a gene regulatory network, which was able to capture biological processes associated with the disease. In addition to the biological insights, this also showed that (a) the cells harvested from donor lungs and cultured for a number of passages keep their regulatory state, potentially encoded as epigenetic memory and (b) our regulatory network approach is able to capture the important biological functions. We note that the gene regulatory network we constructed is based on the interactions between TFs and regulatory elements that are varying across individuals. Thus, we are unlikely to retrieve the highly conserved and tightly controlled processes, but rather those that can vary across individuals and therefore might play a role in converging on the molecular pathways that lead to the disease. We also note that one pitfall of using CTCF-ChIA-PET as a basis for calling chromatin regulatory domains is that we will be blind to any gene loop that is not demarcated by CTCF.

Exploring this regulatory network using disease-specific TFs, we identified processes related to smooth muscle cell differentiation among the top enriched GO terms. This is particularly interesting in the light of our previous finding where we have shown that reduced activity of BMPR2 in PAECs promotes EndMT[26]. However, this previous finding was a specific response to BMPR2 down regulation and mediated by HMGA1 and Slug. What we find here in our global regulatory network analysis, is, that the overall regulatory state potentially drives the cells towards EndMT. This is particularly interesting since we do not find a significant down-regulation of BMPR2. Slug and HMGA1 are target genes in our network and likely contribute to the enrichment of EndMT as a PAH process.

Finally, despite not finding any differentially expressed genes, the H3K27ac levels in PAECs were highly predictive of the expression change in response to endothelial signaling factors for the selected set of genes. This indicates that while these cells, isolated from the innermost endothelial layer of pulmonary arteries still possess endothelial fate, they do show a priming towards a mesenchymal transition into smooth muscle cells. The transition might be triggered by inflammation[30] and by signaling molecules like serotonin and TGFβ. EndMT results in the loss of endothelial factors that repress smooth muscle cell proliferation, and thus contribute to the expanding neointima of proliferating smooth muscle cells that occludes the vessel lumen in PAH patients.

Overall, we present a powerful framework for integrating multiple omics data in enhancer-mediated gene networks that allow the identification of genes driving disease-specific pathways and give insights to disease mechanisms. It shows that steady-state expression analyses are limited in systems where the mechanism involves an aberrant response to stimuli. Considering enhancers, and in particular the active mark H3K27ac, might lead to insights that are not possible with gene expression or promoter analyses alone. It will be interesting to see whether H3K27ac marked enhancers can act as a priming factor in other systems, thus somewhat adding a more dynamic perspective to steady state studies.

## Methods

**Ethics statement**. Procurement of all tissues and cells from human subjects is approved by the Administrative Panel on Human Subjects in Medical Research at Stanford University (IRB#350, Panel 6). Written informed consent was received from participants prior to inclusion in the study. Demographics of Patients and Controls is in Supplementary Table 1.

**Biopsies, cell culturing, and stimulations**. Primary human small pulmonary arterial endothelial cell (PAECs) were harvested from explanted lungs of patients undergoing transplantation for IPAH or HPAH, or from unused donor control subjects, obtained through the Pulmonary Hypertension Breakthrough Initiative. All patients were screened for a panel of PAH genes that include *BMPR2, ACVRL1, ENG, SMAD9, CAV1, KCNK3,* and had none of these mutations except for two *BMPR2* mutation. The *BMPR2* mutation carriers are indicated in Supplementary Data 1. We acknowledge that by the date of publication additional PAH genes have been added to the panel, for which our patients have not been screened (*SMAD1, SMAD4, KCNK3, EIF2AK4, TBX4, ATP13A3, AQP1, SOX17, GDF2, BMP10, KLK1, GCCX*) and we can therefore not exclude that one of the patients has one of these rare mutations. The PAECs were cultured for 3–5 passages in commercial EC media (ECM) supplemented with 5% FBS, penicillin/ streptomycin, and Glutamine (ScienCell), before being harvested for qPCR, RNA-Seq, ChIP-Seq, ChIA-PET or stimulation experiments. As in our previous studies[9,10], we restricted the endothelial cells used in these experiments to passages 3–5 as we have found no evidence of de-differentiation in these early passages. Lack of de-differentiation was judged by a tight cobblestone morphology, prominent CD144 (ve-cadherin) cell boundaries between all cells and uniform uptake of acetylated LDL.

For the growth factor stimulation experiments, cells were starved in ECM basal media with 0.2% FBS for 12 h, before they were treated with serotonin (10 μM), VEGF (10 ng/mL) or TGFβ (10 ng/ml) for 24 h and then harvested for further experiments (qPCR, western blots).

**Reverse-transcriptase quantitative polymerase chain reaction**. Total RNA was extracted and purified from PAECs with the RNAeasy Plus Kit (Qiagen) and then reverse transcribed using SuperScript III (Invitrogen) according to manufacturer's instructions. Expression levels of selected genes were quantified using pre-verified Assays-on-Demand TaqMan primer/probe sets (Applied Biosystems) and normalized to ribosomal RNA 18S. The following primers (Tagman probes from ThermoFisher):

- NOS3: Hs01574665_m1
- HTR1B: Hs00265286_s1
- DKK1: Hs00183740_m1
- TSC22D1: Hs01553787_mH
- RBMS3: Hs01104892_m1
- NFE2L2: Hs00975961_g1
- TGFBR2: Hs00234253_m1
- YAP1: Hs00902712_g1

**ChIP-qPCR**. ChIP-qPCR was measured for three control and three PAH PAECs after 24 h of TGFbeta stimulation (10gn/ml). The following primers were used:

TGFBR2 enhancer_2 chr3:30591954-30592753
TGFBR-E1-B-Fow:TGAGCAGGAAGTTTGGGGAG
TGFBR-E1-B-Rev:CCACTGGTGCAGAGAGACAA
TGFBR2 enhancer_3 chr3:30654771-30655248
TGFBR-E3-A-Fow:CATGCCTATGGGGTGAGGTG
TGFBR-E3-A-Rev:TTCTGGACAGCGAATGTGGG
TGFBR-E3-B-Fow:CCCACATTCGCTGTCCAGAA
TGFBR-E3-B-Rev:AACCCTCGTGCAGAGATTGC
NOS3 enhancer_1 chr7:150689410-150692445
NOS3-E1-A-Fow:GTTTCCCTAGTCCCCCATGC
NOS3-E1-A-Rev:AGAGAGACTAGGGCTGAGGC
NOS3 enhancer_3 chr7:150103672-150104195
NOS3-E3-A-Fow:AGAGCTGCCGCCTTTAATGT
NOS3-E3-A-Rev:CTCAAGGACAGGGGGAGGTA
NOS3-E3-B-Fow:ATGAAGGGAGCTGGGTTTGG
NOS3-E3-B-Rev:GGACTGGTCTCAGCGTGATT
YAP1_enhancer_1: chr11:101980436-101983343
YAP1-E1-A-Fow:TTACAGTCGGCTGAAACGCT
YAP1-E1-A-Rev: CCTCGGTACTGTAAAGCGCA
YAP1-E1-B-Fow:GCCGGCTCACGGTATCTATT
YAP1-E1-B-Rev:GAGAGAGGATGTGCGAACCC
YAP1_enhancer 2: chr11:101917925-101919371
YAP1-E2-A-Fow:CGCCATTCTGAGTGAACCT
YAP1-E2-A-Rev:TCTCAGAGCGCACCGATAAC
YAP1-E2-B-Fow:GTTCGGCTTTGACCAGACCT
YAP1-E2-B-Rev:TCCGACCCCCATACTAACCA

**Study subjects**. Study subjects are described in Supplementary Data 1.

**RNA-Seq profiling.** RNA samples were extracted using the Qiagen All-Prep kit per manufacturer instructions. Libraries were prepared from total RNA using the TruSeq Stranded Total RNA Library Prep Kit per manufacturer instructions. All libraries were sequenced on the Illumina Hiseq 4000.

**ChIP-Seq profiling.** Briefly, Cells were cross-linked with formaldehyde for 10 min (final concentration of 1%). From each sample, 20 million cells were obtained. Chromatin was sheared and subjected to immunoprecipitation with antibodies against H3K27ac, H3K4me1, or H3K4me3. Libraries of the enriched fragments were prepared using Illumina TruSeq library preparation and finally subjected to paired-end sequencing.

In more detail, chromatin immunoprecipitation followed by massively parallel sequencing was carried out as follows and previously described[31]. Cells were cross-linked for 10 min with formaldehyde (final concentration of 1%) at room temperature. Glycine (final concentration of 125 mM) was used to quench the reaction. Nuclear lysates were sonicated using a Branson 250 Sonifier (100% duty cycle for 7 × 30-s intervals; power setting 2). Clarified lysates from 20 million cells per sample were incubated with 1–5 μg of antibody (H3K27ac [Abcam #4729], H3K4me1 [Cell Signaling Technology #5326], and H3K4me3 [Cell Signaling Technology 39751]) coupled to Protein G Dynabeads (Lifetechnologies #10003D, New York). The antibodies have already been used in our previous studies[14,31,32], in which they were validated according to ENCODE standard. RIPA buffer was used to wash the protein-DNA complexes, which were then eluted in 1% SDS TE at 65 °C. After cross-link reversal and purification, the ChIP Illumina DNA Tru-Seq DNA Sample Preparation Kit Instructions (Illumina Part # FC-121-2001, San Diego, CA) were used to generate the sequencing libraries. Pooled libraries were sequenced on an Illumina Hi-Seq 2000 sequencer. Each ChIP-Seq sample was sequenced with 2 × 101 bp paired-end reads at an average depth of ~7.4 million paired reads per dataset (Supplementary Fig. 1b).

**ChIA-PET data generation.** We performed ChIA-PET experiments with modifications to previously published protocols[33,34] and as described in the following: Cells were crosslinked and subjected to nuclear lysis followed by chromatin shearing. Immunoprecipitation was performed overnight at 4 °C with antibodies against CTCF (Cell signaling technology [#3418]). The immuno-complexes were pulled down with Protein-G dynabeads (Life Technologies #10003D, New York). Biotinylated linkers were ligated to the enriched fragments by proximity ligation overnight at 16 °C. Crosslinking was reversed at 65 °C with use of Proteinase K followed by DNA purification. We used Illumina Nextera Transposase to add sequencing adapters to ChIA-PET libraries, thus using Illumina's Nextera tagmentation to generate sequencing libraries, which was different from the previously published protocols[33,34]. Biotinylated fragments were enriched by pull-down with Streptavidin Dynabeads (M-280; Lifetechnologies #11205D, New York). The final libraries were sequenced on the Illumina HiSeq 2000. The data was processed following the Mango toolkit[35].

**RNA-Seq analysis.** The processing and analysis workflow for the RNA-Seq data is shown in Supplementary Fig. 1a. Specifically, RNA-Seq data were subjected to quality control with FastQC before and after adapter trimming. All but one sample passed the quality control. Illumina adapters were trimmed with Trimmomatic V0.33 using paired-end mode and the following parameters: ILLUMINACLIP: TruSeq3-PE.fa:2:30:10 LEADING:3 TRAILING:3 SLIDINGWINDOW:4:15 MIN-LEN:36. GRCh37 assembly from ENSEMBL release 75 were downloaded for alignment and annotation respectively. RNA-Seq data in FASTQ format was mapped using STAR v.2.5.2b (https://github.com/alexdobin/STAR) and gene-level abundances were estimated internally in STAR using the algorithm of HT-Seq (https://pypi. python.org/pypi/HTSeq). Prior to differential expression analysis, lowly expressed genes were filtered out, keeping genes with two read counts or more in at least 10% of the analyzed samples ($n = 18$), and normalized based on size factor estimations. Differential expression analysis was carried out using DESeq2[36] for detecting up- and down-regulated genes in PAH using the following generalized linear model: ~gender+mutation+condition. Subsequent exploratory analysis was conducted using clustering and principal component analysis to assess sample relatedness across metadata and as a quality-control check.

**ChIP-Seq analyses for histone modifications.** The processing and analysis workflow for the ChIP-Seq data is shown in Supplementary Fig. 1a.

Alignment of ChIP-Seq reads: ChIP-Seq data were subjected to quality control with FastQC before and after adapter trimming. Only samples that passed quality control, and had more than 4 million reads, were processed further (see Supplementary Data 1). Illumina adapters were trimmed with Trimmomatic[37] V0.33 using paired-end mode and with these parameters: ILLUMINACLIP: TruSeq3-PE.fa:2:30:10 LEADING:3 TRAILING:3 SLIDINGWINDOW:4:15 MINLEN:36. ChIP-Seq reads of 100 bp in length were aligned to the human genome assembly (hg19) with the Burrows–Wheeler Aligner[38] version: 0.6.1 using "bwa all" and "bwa sampe". Aligned ChIP-Seq data in sam format were transformed to bam files and non-uniquely mapped reads and reads mapping to mitochondrial chromosome were filtered-out using SAMtools. We also removed duplicated reads using Piccard (picard-tools-1.119).

ChIP-Seq peak identification and differential analysis: For peak-calling, we down-sampled all samples that had a library size larger than the median per ChIP-Seq experiment to the median library size. The fragment length of each library was determined using ChIPQC R/Bioconductor package[39]. We called peaks for each sample independently based on the filtered and possibly down-sampled ChIP-Seq data using MACS2 (version: macs2-2.1.0.20150731, with parameters –nomodel –extsize [estimated for each library] –p value 0.01). The number of peaks per individual are shown in Supplementary Table 1. We then removed a consensus of blacklisted regions that are excludable for hg19 in ChIP-Seq experiments (regions obtained from wgEncodeDacMapabilityConsensusExcludable.bed) using BEDtools. A consensus peak set was built for each ChIP-Seq experiment using the dba.peakset function from DiffBind R/Bioconductor package, keeping only peaks that were present in at least two samples. The count matrix where rows represent the consensus peaks and columns are the samples was built using the dba.count function from DiffBind. Exploratory analyses, such as MA-plots, clustering and principal component analysis was carried out on log-transformed counts to assess sample relatedness across metadata and as a quality-control check. To reduce systematic biases induced by high-throughput sequencing, we normalized the ChIP-Seq read count matrix using a cyclic LOESS approach using normOffsets from the csaw package[40] and we removed low-quality samples from downstream analyses. Differential histone modification analyses were carried out using DESeq2[36] for detecting over- and under-modified genomic regions in PAH by employing the following generalized linear model: ~batch+gender+mutation +condition.

**Differential transcription factor activity using diffTF.** The diffTF method[8] is based on the results of our hQTL study where we found that genotype-dependent variation in H3K27ac signal can be explained by disruptions of TF motifs whenever the hQTL-SNP overlapped with a TF binding site[14]. And without knowing the exact mechanism we can assume that TF binding plays an important role in mediating the effect of the DNA variants onto chromatin marks. In diffTF we turn this relationship around and use H3K27ac at putative binding sites of a given TF as a readout of its binding activity. To do so, we obtained the putative binding sites for 601 TF by scanning the genome with the respective position weight matrices (PWMs) using PWMscan (Ambrosini G., PWMTools, http://ccg.vital-it.ch/pwmtools). The PWMs for each TF were obtained from the HOCOMOCO database[41], which collects this information from numerous ChIP-Seq experiments. The TFs were then filtered based on expression level to ensure they are actually expressed in at least two samples. For each TF, diffTF then calculates the average fold-change in chromatin accessibility across all its putative binding sites, and normalizes this value to the average fold-change across the entire genome to estimate its relative binding activity between two conditions (see schematic in Supplementary Fig. 3a). It is important to note that the signal differences at each of the individual putative binding sites is very low and rarely significant, however, by averaging across the thousands of putative binding sites per TF we are able to detect the global differences in binding activity of the TF. TFs with an inconsistent signal for their different motifs were removed these are EGR1, MAFG, RARA, TLX1, TGIF1, VDR, TFDP1. We repeated the same procedure considering H3K27ac peaks that overlapped a TF-specific ChIP-Seq peak in the ChIP-Seq database from ReMap 2018[42].

**Building a multi-omics' gene regulatory network.** We built an enhancer-mediated gene regulatory networks using a multi-omic approach that makes use of co-variation of enhancer activity and gene expression across individuals. Our approach combines expression (RNA-Seq) of TFs and H3K27ac signal (ChIP-Seq) to build TF-enhancer links. Further, links between enhancers and putative target genes are drawn by correlating the H3K27ac signal at enhancers with the expression of target genes that are within CRD that we obtain from cohesin ChIA-PET data. We are using cohesin ChIA-PET to restrict the potential set of enhancers for each gene by the requirement that they have to share a CRD. This is based on the observation presented in Fig. 2a were we found that H3K27ac peaks within cohesin loops were often highly co-regulated.

We follow three main steps to build a PAECs-specific gene regulatory network:
1. Determination of chromatin regulatory domains (CRDs) based on the co-variation of H3K27ac signal at genomic regions enclosed by CTCF loops. This is to restrict the number of potential gene–enhancer pairs.
2. Identification of genes that are regulated by proximal or distal regulatory elements within the same CRD—based on correlation of H3K27ac signal and RNA expression of putative target genes.
3. Identification of the links between TFs and enhancers by correlating TF expression and H3K27ac in enhancers that contain the TF motif.

The three steps are described in more detail in three independent sections below.

In summary, our co-regulation-based approach allowed a multi-omics' integration to define regulatory interactions between TFs and their putative target genes that are mediated by local and distal regulatory elements.

**Defining CRDs.** We detect CRDs in PAECs based on CTCF-mediated chromatin interactions and the co-regulation of their H3K27ac regions. The goal was to

identify cohesin loops that contain highly co-regulated H3K27ac peaks. Since correlation between H3K27ac peaks decreases with distance, we based our detection of CRDs on the assumption that if the most distant pair of H3K27ac peaks within one cohesin loop is correlated, all the peaks within the loop would be equally or stronger correlated. Therefore, we only examined H3K27ac regions at 5Kb up- and down-stream of both anchor points of the chromatin interactions. We then calculated all pairwise Pearson correlations between H3K27ac signal located at opposite anchor points of the interactions (see schematic in Fig. 2c), and since we are studying an active histone modification and expect that distal H3K27ac regions within the same CTCF-mediated interaction will be positively correlated across samples selected only CTCF-mediated interactions, for which H3K27ac peaks at both ends were positively correlated ($P < 0.05$). These were classified as CTCF-mediated CRDs.

**Mapping regulatory elements to their target genes**. We selected gene-H3K27ac peak pairs within the same CRD and defined a regulatory interaction between them if the gene expression and H3K27ac signal of regions within CRDs were positively and significantly ($P < 0.05$) correlated using Pearson correlation.

**Identifying TF-target enhancer links**. We developed a method to define significant regulatory interactions between TFs and H3K27ac regions within CRDs, based on the assumption that TF expression and H3K27ac signal at its target sites are correlated. For estimating the significance of regulatory interactions, we used an empirical estimation of false discovery rate (FDR) that evaluates for each TF the correlation value between its expression (RNA-Seq) and H3K27ac signal (ChIP-Seq) at genomic regions with and without a known TFBS of the given TF (foreground and background respectively). We here defined known binding sites using the same approach described in the TF activity section using the HOCOMOCO database. For any TF, the empirical FDR was then calculated as the number of H3K27ac regions without a TFBS passing the threshold divided by the number of H3K27ac regions without and without a TFBS passing the same threshold. As the local regulation of a TF can be in many cases either activating or repressive, we calculated two empirical FDR curves per TF: one for positive correlation values and another for negative correlation values using increasing or decreasing thresholds, respectively. We set the threshold for 20% FDR for all TFs, which means we used a different correlation threshold for each TF. Finally, we removed interactions that had an absolute correlation (Pearson's coefficient) lower than 0.4 after the FDR threshold selection to remove weak connections.

**GO-term enrichment analysis**. GO enrichments were calculated using the Fisher's exact test and p-values were adjusted using the Benjamini-Hochberg procedure. For each analysis we used the expressed genes as background.

**Reporting summary**. Further information on research design is available in the Nature Research Reporting Summary linked to this article.

## Data availability

The raw sequencing data are available at the Gene Expression Omnibus (GEO) data repository under the accession number GSE126325. Chromatin modifications profiling data have been deposited under the accession number GSE126322. RNA-Seq data have been deposited under the accession number GSE126262. Chromatin interaction profiling (ChIA-PET) data mediated by CTCF have been deposited under the accession number GSE139234. The source data underlying Figs. 1f, 2a, b, d, e, h, 6c and Supplementary Figs. 2a, 4a, b are provided as a Source Data file.

## Code availability

Code is available upon request from the corresponding author judith.zaugg@embl.de.

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

## Acknowledgements

A.R.P. has been recipient of a postdoctoral fellowship granted by Fundación Ramón Areces and the Research Program "Atracción de Talento de la Comunidad de Madrid" (2017-T2/BMD-5532). Our work was supported by K99 HL135258 (M.G.), by the Vera Moulton Wall Center for Pulmonary Vascular Disease (F.G., M.K.), by the National Science Foundation No. 81703877 (S.M.), by the National Basic Research Program of China 973 Program 2015CB554400 (S.M.), by the NIH grants HL122887 (M.R.).

## Author contributions

M.R., M.P.S., and M.K. conceived the experiments. A.R-P. and J.B.Z. conceived the computational analysis. A.R-P. performed the majority of the computational analyses. M.G. and F.G. performed the majority of the experiments. I.B., C.A., and R.S. contributed to the data analysis. S.S., M.S., S.M. contributed to the experiments. M.P.S. and M.R. supervised the experiments, J.B.Z. supervised the computational analyses. J.B.Z. and A.R-P. wrote the paper with help from M.R. All authors read the paper.

## Competing interests

The authors declare no competing interests.
