## [Peer Review File · Nature Communications]

Reviewers' comments:

Reviewer #1 (Remarks to the Author):

Pulmonary arterial hypertension (PAH) is a rare disease which manifests on a very heterogeneous molecular background, very likely also impacted by environmental factors. Reyes-Palomares and coworkers have undertaken a large-scale multi-omics approach to further our understanding of the disrupted gene-regulatory network underlying PAH. Through the integration of several layers of cell type specific 'omics data (i.e. RNA-seq, histone modification (H3K27ac, H3K4me1, H3K4me3) ChIP, chromatin conformation (ChIA-PET)) derived from pulmonary artery endothelial cells (PAEC) the authors conclude that a) the enhancer landscape in PAH is significantly remodelled, b) there is no detectable correlation between these findings and gene expression level, and c) the derived gene-regulatory networks recapitulate already known disease-specific modules as well as processes related to smooth muscle cell differentiation. The presented multi-omics data integration strategy is a promising approach addressing current questions regarding the underlying genetic and genomic etiology in PAH.

Major concerns:

Given the known genetic, molecular and phenotypic heterogeneity of a rare disease like PAH a multi-omics study with just 10 PAH cases (including two BMPR2 mutation carrier (HPAH) and one drug and toxin induced PAH case (APAH)) and 9 controls seems not a large enough sample size to draw any meaningful conclusions from. To rebut this concern the authors would need to provide a substantial power calculation taking the multiple axes of variance and confounding factors into account.

The authors also do not state how the mutation status of the proband and donor lungs have been determined. With the RNA-seq data relevant PAH disease genes could be screened for protein-coding variants or a gene panel test could be used. Furthermore, the definition of PAH disease genes is not clear and needs clarification also for the downstream analysis. The authors are advised to follow the most recent WORLD SYMPOSIUM ON PULMONARY HYPERTENSION (WSPH) 2018 guidelines on genetics and genomics published in the ERJ last year.

The study would also benefit from analysing the observed sequence variation in relevant TFBS for at least a few TFs in data from recent sequencing efforts like gnomAD (<https://www.biorxiv.org/content/10.1101/573378v2>), TopMED (<https://www.biorxiv.org/content/10.1101/563866v1>) and NIHRBR-RD PAH (<https://www.ncbi.nlm.nih.gov/pubmed/29650961>).

Minor concerns:

- There are many syntax errors and typos in the text and figure legends that need attention (see annotations in attached pdf).
- An ethics statement is missing.
- Methods don't seem detailed enough to make the work reproducible and reordering the paragraphs in the sequence the analyses have been performed would be very beneficial.

Reviewer #2 (Remarks to the Author):

In this interesting study, changes in histone marks are examined in diseased and normal human endothelial cells and multi-omics approaches used to work out a chromatin regulatory regime. The

analyses of transcription factor binding and potential activity are novel and noteworthy in that the investigators identify (and report) not only known endothelial transcription factors but also those previously unknown. There are several areas in which key analyses are missing to support the interpretations of the data as shown. Specific comments are as follows:

What was the trend when comparing the three histone marks between hPAH and iPAH (e.g. the comparable analyses as was shown in Figure 1E for control vs PAH)?

Could the lack of difference in RNA-seq between controls and PAH be due to passaging of the cells? What is known based on single cell experiments from other publications or the papers cited (Gu et al., 2017; Rhodes et al., 2015; Sa et al., 2017)? Another concern, returned to below, is how thresholding was performed to determine significant differences in gene expression between health and diseased cells versus the experiments pairing enhancers with genes.

The chromHMM studies are interesting, but is this not a circular argument (methods state: "...we assessed their distribution with respect to annotated chromatin states classes using the chromHMM states for human umbilical vein endothelial cells...")? Heterochromatin states are defined in chromHMM by absence of H3K27ac and H3K4me1/3, such that if one uses that as a definition of the state, H3K27ac will naturally be absent. What do you find if you run chromHMM on your data from patient-derived cells, rather than on HUVECs?

How frequently did the ChIA-PET data reveal enhancers to be looped to the nearest downstream gene? This data should be shown quantitatively in the paper as it provides experimental relevance to the often used method of assuming that enhancers are modulating the nearest downstream gene and is dependent on a ChIA-PET experiment capturing some loops that contain multiple genes, which Supplemental Figure 2 shows did occur in these experiments. The criteria used for CRD definition does not seem to influence the number of genes per loop. What do the results look like if you just assume regulation of nearest downstream gene, completely independent of ChIA-PET data?

Related, how often was the strongest correlation between a changing H3K27ac peak the nearest downstream gene? What was the distribution of H3K27ac peaks per CRD? What was the variability of change in H3K27ac peak when multiple peaks occurred in a CRD?

ChIA-PET takes a chromatin loop or enhancer centric viewpoint, rather than a gene centric viewpoint, whereas it is known that genes can be regulated by multiple enhancers. How do the authors account for this point? Can you use the diffTF package to look for enhancer-gene pairs independent of the ChIA-PET loops?

If the enhancer change correlates positively with the phenotype of the cell and the RNA-seq does not, how can the changing enhancer peak intensity correlate positively with expression of connected genes? Are different thresholds for significant change used between total RNA-seq analyses and the analyses of transcription in the paired analyses with enhancer occupancy?

How do the results in Figure 2F compare to a similar GO analysis on all changing H3K27ac peaks, regardless of ChIA-PET data or CRD determination? What terms are enriched in the total changing H3K27ac dataset?

Figure 4C does not convey a clear message; it is difficult to extract quantitative trends amongst the numerous edges.

This statement seems too speculative, especially in the results section: "Other terms, such as 'cellular response to corticosteroid stimulus' (OR=3.2, adj. p-val=0.02) and 'regulation of vasoconstriction' (OR=3.7, adj. p-val=0.016) might be a consequence of the disease and the treatment the patients obtained prior to their lung transplant (Figure 5A)." However, this is

potentially testable if the investigators have access to medical records. It is again odd that the aspects of this network of enhancer-gene interactions would be maintained through cell culture but transcriptome changes would not, although perhaps this is a general conclusion of the study (i.e. the relative durability of epigenomic states).

Please correct the multiple grammatical errors throughout the paper.

Reviewer #3 (Remarks to the Author):

General comments:

In this study from Reyes-Palomares and colleagues, the authors aimed to address how environmental and epigenetic factors affect the regulatory programmes of disease-specific tissues. To this end, the authors took PAH as a prototype of polygenic disorder, to perform transcriptomic and epigenomic analysis to build a TF-target gene network.

It is an ambitious effort and a challenging read. Environmentally reactive enhancers is a topical concept gaining increased traction in the field but many of the analyses shown here would benefit from more in depth investigation. Several of the findings are of potential interest but lack the support of experimental validation and therefore remain largely speculative. The variation between samples, the limitation of the comparator to healthy controls rather than inclusion of another disease and absence of experimentation that links epigenetic findings to gene expression changes to functional consequences e.g. endoMT in cells weakens the claims of the manuscript. In brief, the specificity and functional link to PAH is lacking. Furthermore, crucial technical aspects should be better explained to the reader and improved to take full profit from these important datasets.

Specific comments

Abstract/introduction

Additional references should be provided to support some bold statements. For example, in line 32, the authors mention that polygenic diseases may arise even in the absence of any risk haplotype. Given that polygenic diseases are generally underlined by many variants with MAF >5% it is rather unlikely that the proposed model is actually observed in real life. Are the authors perhaps referring to a specific feature of the genetic architecture of PAH?

Also, throughout the introduction, the authors give the impression that PAH is a prototypical polygenic disease. Although this may be true, many other polygenic disorders have been much more thoroughly characterised from the genetic point of view, such as Chron's, diabetes, and Alzheimer's to name a few. To the best of my knowledge, there are still very few studies addressing the genetic architecture of PAH and therefore many genetic association signals may still be pending discovery.

The authors postulate that epigenetic mechanisms rather than genetic are responsible for the majority of cases of idiopathic PAH. The role of non-coding elements in PAH has to date been under explored.

Technical aspects

This study relies of PAECs retrieved from PAH and healthy donors. Histological studies of PAH lung at end stage shows marked regional variation. Regional heterogeneity coupled with 3 to 5 passages of cell culture will undoubtedly add variability to the results and the extent to which the observations are reproducible in other hands is an important consideration.

Another is the limitation of the study to PAH (mixed heritable and idiopathic) versus healthy controls. The attribution of changes observed to PAH is unproven.

Figure 1

The authors start by investigating whether endothelial cells in PAH patients show altered transcriptomic and epigenomic profiles, comparing hereditary PAH (n=2), idiopathic PAH and controls.

In this first exploratory analysis, the authors should provide some clarification of the analysis. For example, are the signals plotted in Fig1B the average of all samples, or are they representative data; specifically, is the enrichment consistent across all samples?

To strength the validity of the results, the authors should provide a correlation analysis for signal across all samples/conditions. This analysis will give the reader an appreciation of the dispersion of the signal across samples. The PCAs provided are not enough to make that point. A correlation analysis will also be more relevant than Fig S1B, which could be moved to a supplementary table.

The authors should provide the total number of regions obtained by peak calling with MACS2.1 for each ChIP – this is considerably more informative than simple aligned read counts.

The finding of such pervasive chromatin-level alterations without concomitant changes in gene expression is very puzzling, especially after the authors started this section by stating that they observed a strong correlation between RNA levels and K27ac levels. Ideally, the authors should provide two additional pieces of evidence to support this point: 1) screenshot views of the enrichment signal at the genes analysed by qPCR in Fig1F; 2) in a supplementary figure provide a view of enrichment signals across different replicates.

In addition, Figure S1F shows a substantial number of genes with >1.5 fold-change in PAH vs control, which poses again the question of whether the lack of differentially expressed genes in the presented analysis is real, or simply reflection of large signal variation across the tested samples.

Figure 1F also shows considerable heterogeneity of gene expression and data dispersion. The authors should discuss this as a possible reason for lack of statistically significant differential expressed genes.

Could this variability be technical, reflecting that not all samples were harvested at the same time point? As the authors state, PAECs were kept in culture between 3 and 5 passages. Is it known that the transcriptional and epigenomic profiles of PAECs are stable over such long periods of time? In addition to prolonged exposure to in vitro culture conditions, which will differ substantially from the in vivo niche, different culture times may show different residual effects from intrinsic donor conditions, such as inflammatory and circadian rhythm signals.

Figure 1G shows % of K27ac peaks called, presumably across all PAH samples. This analysis is very difficult to follow as it is and does not allow the reader to grasp the number of regions included in each category, as all are shown as percentage of K27ac regions.

Figure 2

As in the previous figure, the authors should make efforts to provide numbers for the analysed datasets. For example: 1F - How many regions of H3K27ac at TSS are being analysed by GO enrichment, an analysis that is hugely biased by the number of regions taken as input.

The authors make efforts to assign distal regulatory elements to the right target genes in the chromatin of PAECs, using ChIA-PET to map CTCF-CTCF loops. Although valid, the authors should discuss this choice as opposed to alternative methods that would directly link enhancers to promoters, such as Hi-ChIP for H3K27ac. Also it is confusing that the authors state that they performed a ChIA-PET for CTCF in the main text, while describing ChIA-PET for the Cohesin

complex in the

Methods – these are conceptually very different experiments, which is correct?

To give an appreciation of the quality of the data, the authors should show the data for some example loci and show a distribution of CTCF loop size (as in Figure 2A), but in relation to the corresponding TAD. It is difficult to grasp how different these loops are from previously identified TADs.

The authors try to make the point that CTCF-mediated loops act as independent regulatory units. As shown in figure 2D, the authors chose to check correlation of signal strength by taking k27ac peaks at the CTCF anchor points. Why specifically these regions? What evidence is there that CTCF anchor points are more coordinated than internal enhancer/promoter elements for a given loop?

Although the CRD genes identified by the authors represent enrichment in the expected GO categories, caution should be taken in assuming that all enhancer-promoter assignments are bona fide interactions. An experimental validation or additional computational analyses should be provided to strengthen this point. Furthermore, all analyses for CRDs and non-CRDs should be split into actively PAEC-expressed genes and non-expressed genes to avoid biases in the analysis.

Figure 1F would benefit from removal of overlapping categories (for example, positive regulation of endothelial cell proliferation, endothelial cell proliferation and regulation of endothelial cell proliferation must all be represented by roughly the same set of genes). As shown, this representation gives a false impression of many different processes being significantly affected.

Figure 3

The authors then applied diffTF to identify transcription factors that show differential activity across the genome in PAH. Although potentially very useful, this method has not, as of now, been peer reviewed and is only available as preprint in bioRxiv. Therefore, the authors conclusions on this section must be taken with caution.

diffTF analysis identified differential TF binding, which does not necessarily mean that these TFs are differentially active per se. What is the proposed mechanism for such differential activity if their expression does not change (since the authors state that there are few differentially expressed genes)?

Importantly, it is not clear from the manuscript how TF binding sites are identified: is it simply based on TF binding motif mapping, or is there actual ChIP-seq data included? Given the large degree of false-positive rate in motif analysis, this point should be clearer.

Linking back to comments made on previous sections, which mechanism do the authors propose for the detected differential TF activity, when there are no detected changes in the expression of the same TFs (as stated by the authors regarding lack of overall changes in gene expression in PAH samples)?

This highly speculative section would benefit from some experimental validation to demonstrate that indeed at least some of the predicted TFs with differential activity are binding with different affinity in healthy vs PAH.

Similarly with GO analyses shown earlier, TF motifs should be grouped by TF family, as many TFs of the same family have nearly identical binding sequence preferences (such as ATF1/7 or RFX3/4 to name a couple).

Figure 4

This is a very interesting approach to link TFs to their underlying gene networks. However, it seems that the authors are only interested in positive associations between enhancer activity and TF binding. This neglects that a good proportion of TFs can act as repressors of enhancer

elements, in which scenario more binding leads to less enhancer activity and target gene expression.

Figure 5

Similarly to previous sections, the findings may be potentially very interesting but should be corroborated experimentally.

Figure 6

At the end of the study, the authors tried to experimentally validate some of the findings, aiming to demonstrate that PAECs from PAH patients have a different enhancer landscape, which primes a set of genes to respond aberrantly to stimulation.

This set of experiments is key to the authors point, but should be accompanied by ChIP analysis: are the enhancers under question in this figure over-activated in stimulated conditions, or is the so-called priming of enhancers in PAH all that is required for the observed changes in gene expression?

YAP: this particular TF is precisely regulated by phosphorylation. WB analysis would therefore more informative of the activity of YAP showing levels of phospho-YAP.

Minor comments:

- Line 20: "(...) devised a disease-specific enhancer-gene regulatory network based that links transcription factors through enhancers to their target genes." – this would better as "(...) devised a disease-specific enhancer-gene regulatory network that links transcription factors through enhancers to their target genes."
- Supplementary Fig1G – figure legend describes a table, this should be revised.
- Randomised controls should be better explained: for example, which regions were taken, across which genomic interval, and how many times was the shuffling performed.
- Axis in graphics should be more self-explanatory, avoiding for example labels such as "Percentage".
- In a few instances H3K27ac is spelled as H3K37 or other.

Response to reviewers (Reyes-Palomares et al)

Firstly we want to thank all reviewers for their detailed and constructive comments, which, in our opinion have strengthened the manuscript significantly. We have added a statement in the acknowledgements that we are very grateful for the constructive reviewers comments. Below is a short summary of the additional major analyses and experiments we have performed, which is followed by a detailed point-by-point response to each comment of the reviewers. Of note, the manuscript describing the differential TF analysis (diffTF) upon which many of the conclusions are based, is now *in press* with Cell reports and we attached the final version of the manuscript for the reviewers information. In addition to the main manuscript we have uploaded a manuscript with all the edits tracked in red so that reviewers can appreciate all the changes.

I) Summary of additional experiments / analyses addressing the main concerns of the reviewers:

1. We have assessed the power for detecting differentially modified histone mark signals. To do so we have performed extensive power calculations using the R package *ssizeRNA* (version 1.3.1) (Yu et al. 2017), which estimates power based on simulations based on a negative binomial distribution assuming a general linear model and allows for unequal dispersion between conditions. These analyses revealed that our sample size provides decent power given the dispersion of the data. We also run the diffTF analysis for subsampled numbers of individuals and found largely corresponding results independent of the subsets used in the comparisons (**Reviewer Fig. 1** – added to Suppl. Figure 1 and 3).
2. We showed that, despite the lack of significantly differentially expressed genes, the changes in activities for TFs are correlated with their change in expression level, which indicates that there our sample may be to heterogenous for detecting significant changes in RNA. To do so, we have classified TFs into activators and repressors (using diffTF (Berest et al. 2018, accepted in *Cell Reports*)) based on whether their expression correlates positively or negatively with the H3K27ac signal at their putative target sites. This revealed that indeed for TFs classified as activators there is a positive correlation between the fold change in RNA and TF activity between PAH and controls (**Reviewer Fig. 2** – added to Suppl. Figure 3).
3. We have performed a more general characterization of the gene-enhancer architecture. These analyses revealed that 83% (n=6153) of the enhancers (defined as H3K27ac peak regulated by a TF and connected to a target gene) don't connect to their closest gene, which is a surprisingly high number and emphasises the importance of our approach for linking enhancers to their target genes. We further found that the number of enhancers per gene increases when using the CTCF loops, thus showing the importance of using ChIA-PET in our study. Finally, we added some examples of genes in the context of their enclosing ChIA-PET loops (**Reviewer Fig. 3** – added to Figure 2, 4, and Suppl. Figure 2).
4. We assessed whether the active form of YAP1 is also increased in PAH vs controls. To do so we have performed Western Blot analyses of the phosphorylated form of YAP1 (**Reviewer Fig. 4** – added to Suppl. Figure 6)
5. We assessed whether the enhancers that we found primed in PAH show even more differential activity upon stimulation, or whether this initial difference is sufficient for the differential response in their target genes. For this we have performed H3K27ac ChIP-qPCR experiments after stimulation for the enhancers for which we have shown differential expression of their potential

target genes upon stimulation. The results show that overall the enhancers get even more activated upon stimulation indicating that enhancers in PAH patient cells are indeed primed towards a differential response that gets more magnified upon stimulation (Reviewer Fig. 5 – added to Figure 6)

Reviewer Figure 1: A: Power analysis of the differential H3K27ac analysis (added to Suppl. Figure 1 and 3). We employed the R package *ssizeRNA* (version 1.3.1) (Yu et al. 2017), which estimates power based on simulations using a negative binomial distribution and assuming a general linear model. The result indicates that with a sample size of 19, which is what we used in the study, we are already in the plateauing range and approximating 60% power at FDR 0.05. **B:** P-value distributions of differential binding analysis of H3K27ac in PAH vs CONT for the real comparison (red) and for a comparison where the labels of the samples were randomized (red) are shown. The enrichment for small p-values (i.e. signal) is only seen in the real comparison. **C:** Some examples of the most significant and least significant differentially modified H3K27ac peaks are shown. **D:** Power analysis of the differential TF activity analysis (added to Suppl. Figure 3). We chose a subset of samples to perform the diffTF analysis and quantified the fraction of TFs that were recovered as significantly differentially (p-value 0.001). Of the total 10 PAH plus 7 control samples we compared each 2 vs 2, 4 vs 4, 6 vs 6, 7 vs 8 choosing 5-7 runs for each

comparison. The y-axis denotes the fraction of TFs that show a change in TF activity in the same direction as for the full 10 vs 7 data (at p-value <0.001). Note that for each permutation, the original design had to be adjusted to the chosen subsample by removing one or multiple variables that were originally in the design formula: batch, mutation, gender. This is particularly true for the 2 vs 2 group. The small variability for 8 vs 7 and 6 vs 6 can be explained by the fact that for the smaller group, either all (8 vs 7) or all except one (6 vs 6) control samples are taken, thereby not allowing much variability. The only variability then comes from the random selection of 6 / 8 samples from the PAH group.

Reviewer Figure 2

Change in TF activity reflects change in RNA expression level. **A:** Schematic of how the classification into putative activators and repressors is done in *diffTF*. **B** Scatterplots of a TF's differential expression based on RNA-seq (y-axis) and differential TF activity based on *diffTF* results (x-axis) is shown for TFs classified as activators (top) and repressors (bottom). For activators and repressors we found a positive and negative correlation respectively between differential activity and differential expression, thus indicating that the differentially active TFs are indeed slightly differentially expressed, just not reaching significance in our cohort.

Reviewer Figure 3: Characterisation of enhancer-gene interactions. **A-B:** Number of H3K27ac peaks (A) and enhancers (B) assigned per gene. When assigning peaks to genes using CTCF loops (orange) the number of peaks per genes is higher than when simply using the nearest gene association (blue) for each peak. In **A** the results for all H3K27ac peaks are shown. In **B** the results for only enhancers, which are defined as H3K27ac peaks that are modulated by a TF (i.e. correlated with the expression of a TF) and associated with a change in gene expression of the potential target genes, are shown. **C: Majority of enhancers do not link to the nearest gene.** We calculated the percentages of enhancers that are linked to the closest downstream gene vs those that are not linked to the closest gene. Strikingly, the majority of enhancers in our GRN are not linked to gene with the closest TSS. **D-E: Distance dependence of H3K27ac-target gene correlations.** Correlations between H3K27ac signal at

enhancers and level of RNA at target genes is uniformly high even for distal peaks. There is no clear relationship between absolute correlation value and distance between enhancer and promoter (**D**), and distance distributions between nearest gene and CTCF-loop defined enhancers are indistinguishable (**E**). **F: Examples of genes within CRDs.** Six genes (shown in Figure 1F) are shown within the context of their ChIA-PET loops (grey), CRDs (red) and enhancer-promoter interactions based on correlations (green).

Reviewer Figure 4

Westernblots of phosphorylated form of YAP1. A: Raw data of p-YAP at residue Ser127 is show for SPAECs of PAH patients vs controls. Measurements were taken after stimulation with TGF-beta. **B:** Quantification of the signal relative to beta-actin, which served as internal control.

Reviewer Figure 5: ChIP-qPCR analysis for enhancers predicted to prime response in PAH patients. A: Scatterplot as shown in Figure 6 comparing the H3K27ac signal in steady-state with qPCR RNA levels upon stimulation with TGF-beta. **B:** H3K27ac signal at selected enhancers (orange, brown and pink box for NOS3, YP1 and TGFBR2), was measured after TGF-beta stimulation in patient vs controls. All of the enhancers showed a trend in the predicted direction and some of them statistically significantly (n=4 patients, 4 controls).

II) Point-by-point response to reviewer comments

Reviewers' comments:

Reviewer #1 (Remarks to the Author):

Pulmonary arterial hypertension (PAH) is a rare disease which manifests on a very heterogeneous molecular background, very likely also impacted by environmental factors. Reyes-Palomares and coworkers have undertaken a large-scale multi-omics approach to further our understanding of the disrupted gene-regulatory network underlying PAH. Through the integration of several layers of cell type specific 'omics data (i.e. RNA-seq, histone modification (H3K27ac, H3K4me1, H3K4me3) ChIP, chromatin conformation (ChIA-PET)) derived from pulmonary artery endothelial cells (PAEC) the authors conclude that a) the enhancer landscape in PAH is significantly remodelled, b) there is no detectable correlation between these findings and gene expression level, and c) the derived gene-regulatory networks recapitulate already known disease-specific modules as well as processes related to smooth muscle cell differentiation. The presented multi-omics data integration strategy is a promising approach addressing current questions regarding the underlying genetic and genomic etiology in PAH.

Major concerns:

R1.1: Given the known genetic, molecular and phenotypic heterogeneity of a rare disease like PAH a multi-omics study with just 10 PAH cases (including two BMPR2 mutation carrier (HPAH) and one drug and toxin induced PAH case (APAH)) and 9 controls seems not a large enough sample size to draw any meaningful conclusions from. To rebut this concern the authors would need to provide a substantial power calculation taking the multiple axes of variance and confounding factors into account.

Authors' response: We agree with the reviewer that it would be ideal to have a larger dataset. However since it is a rare disease and the lung tissue from which the cells are extracted is even more difficult to obtain it will be difficult to significantly increase the numbers. We thank the reviewer for suggesting a power calculation, which in fact revealed that for the major statistical analyses our sample size provides decent power of 60% at FDR 0.05 (see **Summary of Experiments** and **Reviewer Fig 1A**). This is corroborated by assessing the p-value distribution for the real differential analysis of H3K27ac in PAH vs controls vs the p-value distribution when randomly assigning PAH and control labels (**Reviewer Fig 1B**). Also, as shown for a selection of the most strongly and least strongly differentially modified H3K27ac peaks there is a clear separation between the healthy and control individuals (**Reviewer Fig 1C**). We have also re-run diffTF with different subsets of individuals and found that the sample size we used was very robust for the differentially active TFs (see **Reviewer Fig 1D**). Finally, for all our analyses we are taking known confounding factors into account (i.e. we are always using the batch and gender as known covariates into our models).

Authors' action: We have added a power analysis for the differentially modified H3K27ac peaks (**Reviewer Fig 1A** added as new **SFigure 1D**) and differential TF activity analysis (diffTF) using subsets of patients/control samples (**Reviewer Fig 1D** as new **SFigure 3A**).

R1.2: The authors also do not state how the mutation status of the proband and donor lungs have been determined. With the RNA-seq data relevant PAH disease genes could be screened for protein-coding

variants or a gene panel test could be used. Furthermore, the definition of PAH disease genes is not clear and needs clarification also for the downstream analysis. The authors are advised to follow the most recent WORLD SYMPOSIUM ON PULMONARY HYPERTENSION (WSPH) 2018 guidelines on genetics and genomics published in the ERJ last year.

Authors' response: We thank the reviewer for pointing out that this was not clearly stated in our manuscript. In fact, all patients were screened for a panel of PAH genes that include *BMPR2*, *ACVRL1*, *ENG*, *SMAD9*, *CAV1*, *KCNK3* and had none of these mutations except for two *BMPR2* mutation carriers that were indicated in Table 1.

Authors' action: We have added the following statement to the methods: *"All patients were screened for a panel of PAH genes that include BMPR2, ACVRL1, ENG, SMAD9, CAV1, KCNK3 and had none of these mutations except for two BMPR2 mutation. The BMPR2 carriers are indicated in Suppl. Table 1."*

R1.3: The study would also benefit from analysing the observed sequence variation in relevant TFBS for at least a few TFs in data from recent sequencing efforts like gnomAD (<https://www.biorxiv.org/content/10.1101/573378v2>), TopMED (<https://www.biorxiv.org/content/10.1101/563866v1>) and NIHRBR-RD PAH (<https://www.ncbi.nlm.nih.gov/pubmed/29650961>).

Authors' Response: While we agree with the reviewer that this is an interesting analysis we believe that this is out of scope of the submitted manuscript, which mainly focuses on the epigenetic landscapes of PAH patients. Since these are all rare mutations we would not expect that they are present in our small cohort of patients and indeed it would be very difficult to infer any causal mechanism based on variants that we might find in a small number of patients. It would indeed be very interesting to perform a targeted analysis in binding sites of the TFs that we identified in the data deposited by these recent sequencing efforts, which should be done in a next study.

R1.4: Minor concerns:

- There are many syntax errors and typos in the text and figure legends that need attention (see annotations in attached pdf).

Authors' Response and Action: We thank the reviewer for pointing these out and have corrected them (and others - indicated in red in the main manuscript).

- An ethics statement is missing.

Authors' Response: We thank the reviewer for pointing this out.

Authors' Action: We added the following Ethics statement to the methods section: *"Procurement of all tissues and cells from human subjects is approved by the Administrative Panel on Human Subjects in Medical Research at Stanford University (IRB#350, Panel 6). Written informed consent was received from participants prior to inclusion in the study. Demographics of Patients and Controls is in Suppl. Table 1."*

- Methods don't seem detailed enough to make the work reproducible and reordering the paragraphs in the sequence the analyses have been performed would be very beneficial.

Authors Response: We thank the reviewer for this suggestion. We have reordered the methods paragraphs to match with the sequence in which they have been performed.

Reviewer #2 (Remarks to the Author):

In this interesting study, changes in histone marks are examined in diseased and normal human endothelial cells and multi-omics approaches used to work out a chromatin regulatory regime. The analyses of transcription factor binding and potential activity are novel and noteworthy in that the investigators identify (and report) not only known endothelial transcription factors but also those previously unknown. There are several areas in which key analyses are missing to support the interpretations of the data as shown. Specific comments are as follows:

R2.1: What was the trend when comparing the three histone marks between hPAH and iPAH (e.g. the comparable analyses as was shown in Figure 1E for control vs PAH)?

Authors Response': We agree with the reviewer that this is an interesting analysis in principle. While we are happy to report these trends to the reviewer we did not include any analysis comparing hPAH vs iPAH into the manuscript since we had only 2 patients classified as hPAH, which does not allow us to draw strong conclusions about the findings. Nevertheless, we have now performed these analyses and compared hPAH and iPAH for the three histone marks and RNA. We identified 2, 14, and 4 differentially accessible peaks for H3K27ac, H3K4me3 and H3K4me1 respectively at FDR 10% (**Reviewer Fig. 6**). Two of the top 10 most differentially modified regions across all histone marks fell into interesting regions:

- Chr18:25755252-25758739 and chr18:25755174-25758634: up in iPAH for all three chromatin marks. This is a peak in close proximity to CHD2
- Chr2:203244026-203244476: up iPHA for H3K4me3, a peak close to BMPR2, indicating that there could be an epigenetic effect in addition to the mutation in BMPR2

However, while these examples look interesting and may represent real differences between the disease subtypes, we do not want to draw any conclusions based on a comparison in which one group (the hPAH) involves only two patients. We have therefore decided not to add these analyses to the manuscript.

Reviewer Fig. 6: MA plot for iPAH vs hPAH for the three histone marks and RNA. There is almost no differentially modified peaks nor differentially expressed RNA at an FDR of 0.05, which may also be due to the very small sample size for the hPAH cases (n=2 for hPAH and n=7 for iPAH).

Reviewer Figure 6

R2.2: Could the lack of difference in RNA-seq between controls and PAH be due to passaging of the cells? What is known based on single cell experiments from other publications or the papers cited (Gu et al., 2017; Rhodes et al., 2015; Sa et al., 2017)? Another concern, returned to below, is how thresholding was performed to determine significant differences in gene expression between health and diseased cells versus the experiments pairing enhancers with genes.

Author's response: We thank the reviewer for this suggestion. We were ourselves surprised by the lack of significant changes in gene expression in this cohort when compared to our previously studied cohorts (Rhodes et al., 2015; Sa et al., 2017) since in all these studies there was overlap in the patient samples. The passage number was the same as in those studies, so it is unlikely that the passage number has an effect on the result. It is likely that the heterogeneity between patients in our study was greater than in other studies and thus we did not obtain statistical significance. In fact when comparing the correlation coefficients between control samples with the correlation coefficients of patient samples we find lower correlation values within the patients group (**Reviewer Fig. 9**, see also response to R3.7), indicating that there is more variability within the patient group than within the controls, which may be the reason we do not find any statistically significant differences. Notably, there are differences in gene expression (see also **Reviewer Fig. 2**), however they fail to reach statistical significance at 10% FDR.

With regards to the pairing enhancers with genes, this was not based on differential expression but based on co-variation across individuals that may or may not be related to the disease status of the individual (see more detailed explanation in response to **R2.7**).

R2.3: The chromHMM studies are interesting, but is this not a circular argument (methods state: "...we assessed their distribution with respect to annotated chromatin states classes using the chromHMM states for human umbilical vein endothelial cells...")? Heterochromatin states are defined in chromHMM by absence of H3K27ac and H3K4me1/3, such that if one uses that as a definition of the state, H3K27ac will naturally be absent. What do you find if you run chromHMM on your data from patient-derived cells, rather than on HUVECs?

Authors' response: We thank the reviewer for this comment, which indicates that our rationale for performing the chromHMM studies did not come across clearly. As suggested by the reviewers, we do expect H3K27ac regions to be mainly located in active elements and were thus happy to see these enrichments since they serve as validation of the data. What was more surprising was that we found the differentially modified H3K27ac regions even more enriched at enhancers and less enriched at promoters with respect to the full set of H3K27ac peaks. From this we concluded that the H3K27ac remodeling in PAH was mainly happening at enhancers.

Authors' action: We have made the message of the chromHMM analysis more clear and modified the following sentence: "*The differentially modified regions were strongly enriched in active enhancers (adjusted p-value (adj. p-val)=1.7E-278; Odds Ratio (OR)=4.4) and slightly depleted from active promoters (adj. p-val=0.03; OR=0.93) when using all H3K27ac peaks as a background with respect to all non-differentially modified H3K27ac peaks (Figure 1G).*"

R2.4: How frequently did the ChIA-PET data reveal enhancers to be looped to the nearest downstream gene? This data should be shown quantitatively in the paper as it provides experimental relevance to the often used method of assuming that enhancers are modulating the nearest downstream gene and is dependent on a ChIA-PET experiment capturing some loops that contain multiple genes, which Supplemental Figure 2 shows did occur in these experiments. The criteria used for CRD definition does

not seem to influence the number of genes per loop. What do the results look like if you just assume regulation of nearest downstream gene, completely independent of ChIA-PET data?

Author's response: We agree with the reviewer that it is interesting to report the number of enhancers that don't connect to the closest gene in network. Doing so, we found that 80% of the enhancers in our network skip at least one gene, which is a surprisingly high number (**Reviewer Fig. 3C**).

Regarding the CRD definition: while it does not influence the number of genes per loop, it does dramatically increase the biological signal in terms of GO terms enriched in target genes: we get a good set of cell-type specific GO terms enriched for all genes linked to enhancers within our CRDs (3188) while we find no GO terms enriched in an analysis that includes all genes and enhancers that were not in CRDs (3626 - genes in a loop that is not a CRD; **Reviewer Fig. 7**). In addition, it also increases the number of enhancers linked to a gene when compared to simply the closest gene per enhancer (**Reviewer Fig. 3B**).

If we use the closest gene per peak (25294 genes), we cannot calculate any GO enrichment because almost all genes are in the test set. If we select genes connected to differentially modified peaks within CRDs (470) we get a small set of very specific GO terms related to the pathology of PAH (as shown in main **Figure 2F**), while for the nearest genes to differentially modified peaks (6601 genes) we get a very large set of cell-type specific GO terms not necessarily linked to the disease (**Reviewer Table 1**). In summary: CRDs increase the number of enhancers per gene and the genes within CRDs carry more relevant biological signal as measured by GO term enrichment.

Authors action: We have added **Reviewer Fig. 3C** as new **Figure 2C**, and added the following sentences: *“Interestingly, about 80% of the enhancers skip at least one gene, thus illustrating the importance of using CRDs for identifying real enhancer-gene links. Using the CRD criteria we identified significantly more enhancers per gene than if we just take the closest gene per enhancer.”*

Reviewer Figure 7

Rev. Fig. 7: Top GO terms enriched using genes that are within a CRD (full list in **Reviewer Table 1**). A large set of terms is enriched within the 3188 genes within CRDs whereas no single term was enriched among genes that were in non-CRDs CTCF loops (3626).

R2.5: Related, how often was the strongest correlation between a changing

H3K27ac peak the nearest downstream gene? What was the distribution of H3K27ac peaks per CRD? What was the variability of change in H3K27ac peak when multiple peaks occurred in a CRD?

Author's response: We agree with the reviewer that these are interesting explorations. Additional analyses showed that the correlation of H3K27ac peaks was generally not decreasing with distance, which suggests that distance plays a minor role once we filter for gene-enhancer interactions within CRDs (see **Reviewer Fig. 3D**). The mean number of H3K27ac enhancers per gene within CRDs is higher than when we just map enhancers to their closest gene (2.8 vs 1.6 enhancers per gene). The H3K27ac signal within a ChIA-PET loop was typically very consistent (either higher in patients or higher in controls) as shown in **Fig. 2A** of the manuscript.

Authors action: We have added **Reviewer Fig 3C** as Figure 4C and added the following text: *"Interestingly, about 80% of the enhancers skip at least one gene, thus illustrating the importance of using CRDs for identifying real enhancer-gene links. Using the CRD criteria we identified significantly more enhancers per gene than if we just take the closest gene per enhancer."* We have also added a panel about the number of H3K27ac peaks per CRD (**Reviewer Fig. 3B**) to Suppl. Figure 2B: Legend: *"Number of enhancers per gene when using CRDs (orange) vs non-CRD loops (blue)."*

R2.6: ChIA-PET takes a chromatin loop or enhancer centric viewpoint, rather than a gene centric viewpoint, whereas it is known that genes can be regulated by multiple enhancers. How do the authors account for this point? Can you use the diffTF package to look for enhancer-gene pairs independent of the ChIA-PET loops?

Authors response: This comment points out that our description of the CRDs was not clear enough and we apologize for this. The way we defined enhancer-promoter interactions is by taking any peak and gene located within the same CRD and then correlated the H3K27ac signal of the enhancer with the RNA level of the gene across individuals. We chose this approach because the H3K27ac peaks within CRDs tended to co-vary more than expected and we therefore treated the loop as a regulatory unit (similar as has been described in (Delaneau et al. 2019)). Thus, genes can be regulated by multiple enhancers. In fact, an additional analysis showed that over 57% of genes are linked to more than one H3K27ac peak and over 35% are linked to more than two peaks while only 3.7% are linked to ten or more peaks (**Reviewer Fig. 3B**). diffTF does not provide enhancer-gene pairs, it only estimates differences in transcription factor activities.

Authors action: We have clarified the definition of CRDs by adding the following part in the sentence: *"To link genes to their regulatory elements (enhancers, promoters) we calculated the Pearson correlations between the H3K27ac signal and RNA expression of each gene-peak pair within each CRD (i.e. assuming genes are within the same regulatory unit with their enhancers), and defined positive correlations (at a nominal p-value of 0.05) as regulatory (enhancer-gene and promoter-gene) interactions (Figure 2C)".*

R2.7: If the enhancer change correlates positively with the phenotype of the cell and the RNA-seq does not, how can the changing enhancer peak intensity correlate positively with expression of connected genes? Are different thresholds for significant change used between total RNA-seq analyses and the analyses of transcription in the paired analyses with enhancer occupancy?

Author's response: We thank the reviewer for raising this question, since it shows that we have not clearly explained the conceptual differences between differential expression analysis and the generation of the gene regulatory network. For the differential expression we are comparing RNA expression levels between healthy controls and PAH patients and the reason we are not seeing any signal is likely due high variation among individuals and low fold changes between the groups. For linking enhancers to target genes, this high inter-individual variation is very useful - as long as it is biological and

reproducible across the different assays - because the gene-enhancer linking is based on correlating the enhancer signal (H3K27ac) and gene expression (RNA) across individuals regardless of their disease status. Thus it is not different thresholds that were used but conceptually different analyses.

Authors action: We have added the following sentence to clarify the conceptual explanation for the enhancer-gene links: *"Note that this approach is based on the co-variation of TF expression and enhancer signal across individuals and independent of the differential expression/signal between conditions."*

R2.8: How do the results in Figure 2F compare to a similar GO analysis on all changing H3K27ac peaks, regardless of ChIA-PET data or CRD determination? What terms are enriched in the total changing H3K27ac dataset?

Author's response: We agree with the reviewer that such an analysis is important to benchmark the CRDs against. In fact we had done that and decided to only show the analysis of all differentially H3K27ac modified peaks to that fall into promoter regions to make the point about the importance of enhancers. This analysis is described in Suppl. Figure 2A and showed that there are only very few and general GO terms enriched. When we performed a GO enrichment analysis on all differentially modified H3K27ac peaks linked to their closest gene we found many tissue-specific terms enriched that were not necessarily related to the disease (see **Reviewer Suppl. Table 2**), yet the number of genes that went into the analysis was very large (6601). See also response to **R2.5**.

R2.9: Figure 4C is does not convey a clear message; it is difficult to extract quantitative trends amongst the numerous edges.

Author's response: We thank the reviewer for this feedback. The message we wanted to convey with Figure 4C was that there are extensive changes in the gene regulatory network between PAH and control. To convey this message more clearly we have now added the number of nodes and edges in each category.

Authors action: We have updated Figure 4C to include the numbers of edges and nodes.

R2.10: This statement seems too speculative, especially in the results section: "Other terms, such as 'cellular response to corticosteroid stimulus' (OR=3.2, adj. p-val=0.02) and 'regulation of vasoconstriction' (OR=3.7, adj. p-val=0.016) might be a consequence of the disease and the treatment the patients obtained prior to their lung transplant (Figure 5A)." However, this is potentially testable if the investigators have access to medical records. It is again odd that the aspects of this network of enhancer-gene interactions would be maintained through cell culture but transcriptome changes would not, although perhaps this is a general conclusion of the study (i.e. the relative durability of epigenomic states).

Authors response: While we agree with the reviewer that this particular statement might be too speculative, we believe that our study provides indeed strong evidence that the cells keep their "epigenetic" state even in culture. The reason we are not seeing the transcriptome changes doesn't mean they are not present, it just means that they are not significant - likely due to high heterogeneity among the individuals. In fact, we do find that the TFs that are differentially active between patients and controls do also vary on their transcriptional level in the expected direction (see also Additional Analysis and **Reviewer Fig. 2**).

With respect to the medical records, we agree this is an interesting analysis and indeed revealed that all the patients were treated with different kinds of vasodilator drugs that presumably regulate vasoconstriction (combinations of epoprostenol, ambrisentan, sildenafil, sitaxentan, tresprostinil,

bosentan, iloprost, tadalafil, iloprost, see **STable 1** in the manuscript). Most patients are also treated with a prostaglandin (epoprostenol) or prostacyclin analogue (tresprostinil), which are antagonists of corticosteroids.

Authors action: We have added the information about the drugs to the sentence mentioned by the reviewer: “[...] *might be a consequence of the disease and the treatment the patients obtained prior to their lung transplant since all patients were treated with some kind of vasodilator drug (Figure 5A, also see STable 1 for patient treatments).*”

R2.11: Please correct the multiple grammatical errors throughout the paper.

Authors response: We apologise for the grammatical errors and have corrected them.

Reviewer #3 (Remarks to the Author):

General comments:

In this study from Reyes-Palomares and colleagues, the authors aimed to address how environmental and epigenetic factors affect the regulatory programmes of disease-specific tissues. To this end, the authors took PAH as a prototype of polygenic disorder, to perform transcriptomic and epigenomic analysis to build a TF-target gene network.

It is an ambitious effort and a challenging read. Environmentally reactive enhancers is a topical concept gaining increased traction in the field but many of the analyses shown here would benefit from more in depth investigation. Several of the findings are of potential interest but lack the support of experimental validation and therefore remain largely speculative. The variation between samples, the limitation of the comparator to healthy controls rather than inclusion of another disease and absence of experimentation that links epigenetic findings to gene expression changes to functional consequences e.g. endoMT in cells weakens the claims of the manuscript. In brief, the specificity and functional link to PAH is lacking. Furthermore, crucial technical aspects should be better explained to the reader and improved to take full profit from these important datasets.

Specific comments

R3.1: Abstract/introduction

Additional references should be provided to support some bold statements. For example, in line 32, the authors mention that polygenic diseases may arise even in the absence of any risk haplotype. Given that polygenic diseases are generally underlined by many variants with MAF >5% it is rather unlikely that the proposed model is actually observed in real life. Are the authors perhaps referring to a specific feature of the genetic architecture of PAH?

Authors' response: We agree with the reviewer that the absence of evidence is not equal to evidence of absence. In fact we mention that we mean *known* risk haplotypes in the sentence cited by the reviewer. However, we believe PAH is a polygenic disease since BMPR2 mutations account for about 70% of familial cases and 25-30% of sporadic cases of PAH. A large collaborative European cohort of over 1000 patients with IPAH, FPAH and anorexigen associated PAH confirmed causal mutations in BMPR2 (15%), TBX4 (1.3%), ACVRL1 (0.9%), SMAD9 (0.4%), KCNK3 (0.4%), ENG (0.6%). Mutations in CAV1 were not found owing to their rarity. The same study identified new mutations in ATP13A3 (1.1%) SOX17 (0.9%), GDF2 (BMP9) (0.8%), AQP1 (0.9%) (Morrell et al. 2019). Thus there are certainly many genes potentially causally involved in PAH. Nevertheless, we agree with the reviewer that it may be more than just genetic variants and we therefore rephrased the introduction to call PAH a “complex multifactorial” disease.

Authors action: We have added a reference to recent study showing the polygenic causes of PAH (Morrell et al. 2019). We have also added the following sentence to the introduction: *“A large collaborative European cohort of over 1000 patients with IPAH, FPAH and anorexigen associated PAH confirmed causal mutations in BMPR2 (15%), TBX4 (1.3%), ACVRL1 (0.9%), SMAD9 (0.4%), KCNK3 (0.4%), ENG (0.6%) (Morrell et al. 2019). Mutations in CAV1 were not found owing to their rarity. The same study identified new mutations in ATP13A3 (1.1%) SOX17 (0.9%), GDF2 (BMP9) (0.8%), AQP1 (0.9%) (Morrell et al. 2019).”* We have also rephrased the following sentence: *“Here we use pulmonary arterial hypertension (PAH) as an example of a polygenic complex multifactorial disease (Morrell et al. 2019), for which the disease cell type is well known, to examine the contribution of epigenetic alterations in primary cells with the aim of deriving insights into the disease mechanism.”*

R3.2: Also, throughout the introduction, the authors give the impression that PAH is a prototypical polygenic disease. Although this may be true, many other polygenic disorders have been much more thoroughly characterised from the genetic point of view, such as Chron’s, diabetes, and Alzheimer’s to name a few. To the best of my knowledge, there are still very few studies addressing the genetic architecture of PAH and therefore many genetic association signals may still be pending discovery.

Authors’ response: We agree with the reviewer that there are other polygenic diseases that have been investigated in much more detail, mostly because they are much more common than PAH. We also agree that there might be more genetic associations still to be discovered, however we do not see how this will affect the conclusions of our study.

R3.3: The authors postulate that epigenetic mechanisms rather than genetic are responsible for the majority of cases of idiopathic PAH. The role of non-coding elements in PAH has to date been under explored.

Authors’ response: We agree with this statement of the reviewer.

Technical aspects

R3.4: This study relies of PAECs retrieved from PAH and healthy donors. Histological studies of PAH lung at end stage shows marked regional variation. Regional heterogeneity coupled with 3 to 5 passages of cell culture will undoubtedly add variability to the results and the extent to which the observations are reproducible in other hands is an important consideration.

Authors’ response: While we agree that regional heterogeneity and passage may account for some variability, given the similar abnormalities in functional studies with these cells (Sa et al. 2017) and the experimental validations descrscribed in Figure 6, we expect that the observations are reproducible.

R3.5: Another is the limitation of the study to PAH (mixed heritable and idiopathic) versus healthy controls. The attribution of changes observed to PAH is unproven.

Authors response: We acknowledge that our set of patients is heterogeneous and this might explain the lack of signal from on the RNA level. Yet, despite the mixed heritable and idiopathic PAH vs. controls we expect that the changes observed do indeed reflect the disease as this was tested in Figure 6 with observations that were functionally relevant to the known pathogenesis of PAH.

Figure 1

The authors start by investigating whether endothelial cells in PAH patients show altered transcriptomic and epigenomic profiles, comparing hereditary PAH (n=2), idiopathic PAH and controls.

R3.6: In this first exploratory analysis, the authors should provide some clarification of the analysis. For example, are the signals plotted in Fig1B the average of all samples, or are they representative data; specifically, is the enrichment consistent across all samples?

Authors response: We agree with the reviewer that this should be clarified. The data in Fig1B represent the average across all samples.

Authors action: We have updated the description of Fig1B in the legend: "...The normalized read counts (mean across all individuals) for each histone are colored according to their specific legend."

R3.7: To strength the validity of the results, the authors should provide a correlation analysis for signal across all samples/conditions. This analysis will give the reader an appreciation of the dispersion of the signal across samples. The PCAs provided are not enough to make that point. A correlation analysis will also be more relevant than Fig S1B, which could be moved to a supplementary table.

Author's response: We agree with the reviewer that this is an insightful analysis to add. When comparing the global correlations between all individuals we observed similar trends as with the PCA: for the H3K27ac marks the patients were stronger correlated among themselves and distinct from the controls, while no clear difference was visible for the other marks and RNA (see **Reviewer Fig. 8**).

Authors action: We have added **Reviewer Figure 8** as panel in Suppl. Figure 1G and added a statement about it in the main text: "Reassuringly, when we compared the pairwise correlations of the histone or RNA signal between any two individuals we found stronger correlations between pairs of controls than between pairs of patients for the enhancer mark (H3K27ac), the promoter mark (H3K4me3) and RNA (SFigure 1G). This indicates that the lack of significantly differential signal in the promoter and gene expression might be due to the variability of these molecular markers within the group of patients."

Reviewer Figure 8

Reviewer Figure 8 (added to Suppl. Figure 1G): Density plots showing the distribution of correlation coefficients comparing signal across peaks between any pair of individuals stratified by disease status are shown for H3K4me3 (**A**), H3K4me1 (**B**), H3K27ac (**C**) and RNA (**D**). Correlation coefficients are highest for pairs of the control group, indicating that there is little variation within controls (red). Variation within patients (blue) is already significantly higher than within controls, indicating that there is a lot of heterogeneity between patients. As expected, correlation is lowest between pairs of

individuals that include one patient and one control (grey). The exception for this is the enhancer mark H3K4me1, suggesting that enhancers are similar between patients and controls, and that it is really the activity of the enhancers (H3K27ac) and promoters (H3K4me3) or gene expression (RNA) that shows more heterogeneity.

The authors should provide the total number of regions obtained by peak calling with MACS2.1 for each CHIP – this is considerably more informative than simple aligned read counts.

Author’s response: We agree with the reviewer that these are useful numbers to compare and have added them as **Suppl. Table 3**.

Authors action: We have added **Suppl. Table 3** that provides the number of peaks per sample.

R3.8: The finding of such pervasive chromatin-level alterations without concomitant changes in gene expression is very puzzling, especially after the authors started this section by stating that they observed a strong correlation between RNA levels and K27ac levels. Ideally, the authors should provide two additional pieces of evidence to support this point: 1) screenshot views of the enrichment signal at the genes analysed by qPCR in Fig1F; 2) in a supplementary figure provide a view of enrichment signals across different replicates.

Authors response: We agree that at first it was also very surprising for us to see such pervasive chromatin changes without any significant changes in gene expression. As mentioned earlier though, the fact that we don’t detect significant changes in expression doesn’t mean there are no changes (see response to R3.1). In fact, in **Reviewer Fig. 3B** we show that there is a general agreement of the differential expression of TFs and their differential TF activities between patients and controls.

With regards to the suggestions of showing the signal of the enhancers for the genes assayed in Fig. 1F, we agree, this is a great suggestion and we have added an extra panel to Figure 1 (**Reviewer Figure 9**). This also addresses the second suggestion because our “replicates” are in fact the individual patients/controls.

Authors action: We have added Reviewer Fig 9 to **Figure 1G** to shows the signal of the H3K27ac peaks at the genes that were assayed by qPCR. “G: H3K27ac signal at enhancers of the genes assayed by qPCR in F.”

Reviewer Figure 9

Reviewer Figure 9: H3K27ac signal at enhancers of the genes assayed by qPCR in Figure 1F.

R3.9: In addition, Figure S1F shows a substantial number of genes with >1.5 fold-change in PAH vs control, which poses again the question of whether the lack of differentially expressed genes in the presented analysis is real, or simply reflection of large signal variation across the tested samples.

Authors response: We apologize for the mis-labeling and incorrect legend of SFigure 1F, which in fact is displaying the differential H3K27ac signal for iPAH vs controls against the differential H3K27ac signal of hPAH vs controls. From this figure we concluded that the BMPR2 mutation (hPAH patients) does not have a dramatically different effects from the iPAH patients when compared to controls.

Nevertheless, we agree with the statement of the reviewer that it is likely that the lack of RNA signal is due to heterogeneity among the individuals, see also our response to **R3.8** above.

Author's action: We have clarified in the main text that SFigure 1F compares differential RNA expression of HPAH vs controls: "*Similar ~~fold-changes-results~~ were ~~observed~~ ~~obtained~~ when comparing the HPAH patients against controls (SFigure 1F) and we therefore combined the IPAH and HPAH patients into one set for all subsequent analyses*"

Figure 1F also shows considerable heterogeneity of gene expression and data dispersion. The authors should discuss this as a possible reason for lack of statistically significant differential expressed genes.

Authors response: We agree with the reviewer about this. See our response and action to **R3.8**.

R3.10: Could this variability be technical, reflecting that not all samples were harvested at the same time point? As the authors state, PAECs were kept in culture between 3 and 5 passages. Is it known that the transcriptional and epigenomic profiles of PAECs are stable over such long periods of time? In addition to prolonged exposure to in vitro culture conditions, which will differ substantially from the in vivo niche, different culture times may show different residual effects from intrinsic donor conditions, such as inflammatory and circadian rhythm signals.

Authors response: These points are all well taken, and we suggest that the epigenomic profiles may in fact be more stable in these primary cells than the transcriptomic profiles that require stimulation of the cells as shown in Figure 6.

R3.11: Figure 1G shows % of K27ac peaks called, presumably across all PAH samples. This analysis is very difficult to follow as it is and does not allow the reader to grasp the number of regions included in each category, as all are shown as percentage of K27ac regions.

Author's response: We thank the reviewer for this comment, which allows us to clarify this analysis. Indeed these are the consensus set of peaks that we have obtained by merging the peaks across all individuals. We are displaying the percentages of each set of features overlapping peaks so the reader can better appreciate the differences between the original set of Chromatin states from HUVEC and captured by H3K27ac in PAEC since the numbers are very different (515,807 for HUVEC, 263,910 for all sPAECs peaks and 31,084 for differentially modified PAH peaks). However, we agree it is useful to have the number and we now indicate the number of peaks in the legend on the panel.

Authors action: We have updated the legend for old Fig 1G (new Fig 1H) and added the number of peaks in each category: "*Distribution of chromatin states in HUVEC (grey bars, representing 515,807 genomic features) within H3K27ac regions detected in PAECs (blue bars, representing 263,910 genomic features) and in differentially modified regions (orange bars, representing 31,084 genomic features)....*"

Figure 2

R3.12: As in the previous figure, the authors should make efforts to provide numbers for the analysed datasets. For example: 1F - How many regions of H3K27ac at TSS are being analysed by GO enrichment, an analysis that is hugely biased by the number of regions taken as input.

Author's response: We agree with the reviewers that these are important numbers to consider and we have added them to the revised manuscript. In this case the number of tested genes is 470. This number is referred to Figure 2F (instead of Figure 1F) for genes that are linked to differentially acetylated regions within CRDs that are not necessarily located at gene TSSs. In case of the SFigure 2A, we tested 1,301 genes showing differentially modified regions at their promoters, defined as 3.5kb up- and downstream of their TSS.

Authors action: We have added the number of genes that were used for calculating the GO enrichments in Figure 2F.

R3.13: The authors make efforts to assign distal regulatory elements to the right target genes in the chromatin of PAECs, using ChIA-PET to map CTCF-CTCF loops. Although valid, the authors should discuss this choice as opposed to alternative methods that would directly link enhancers to promoters, such as Hi-ChIP for H3K27ac. Also it is confusing that the authors state that they performed a ChIA-PET for CTCF in the main text, while describing ChIA-PET for the Cohesin complex in the Methods – these are conceptually very different experiments, which is correct?

To give an appreciation of the quality of the data, the authors should show the data for some example loci and show a distribution of CTCF loop size (as in Figure 2A), but in relation to the corresponding TAD. It is difficult to grasp how different these loops are from previously identified TADs.

Authors response: We apologize for the mistake in the description of the ChIA-PET data (which was mistakenly taken as reference from another manuscript we have been working on at the same time). We used CTCF for the IP and have now updated the description.

Regarding the differences between loops and TADs: it is mainly the size that is very different between these structures (albeit the differences may be more philosophical than quantitative). TADs tend to be much larger (in the megabase range) than loops (in the kilobase range). While it might be interesting to compare loops and TADs we believe that this is outside the scope of our study since we are mainly using the loops for defining gene-enhancer interactions, yet we agree that it will be nice to show a couple of examples for the ChIA-PET data and CRDs and we have added them now to the Figures (**Reviewer Figure 3** added to Figure 2 and SFigure 2). The data quality and a more comprehensive analysis of the ChIA-PET loops - including the ones that we are using in this study, will be described in a companion paper of the current round of ENCODE papers and is accepted at Nature. In **Reviewer Fig. 10** we show the quality in terms of the RSC and NSC in relation to the ENCODE ChIA-PET that was recently accepted at Nature. Our two replicates fall well above the ENCODE standard and well within the distribution of ChIA-PET data from ENCODE.

Author's action: We have added a quality control plot that shows the quality of our CTCF ChIA-PET data relative to the RAD21 ChIA-PET data that is part of ENCODE and got recently accepted at Nature. Specifically, we added Reviewer Fig. 10 to Suppl. Figure 2B and the examples in Reviewer Figure 1F to Figure 2G and SFigure 2E. We added the following text to the manuscript: "*The data quality was above ENCODE standards and in a similar range to ENCODE ChIA-PET (SFigure 2B)*" and "*Some examples of genes within CRDs are shown in Figure 2G and SFigure 2E.*"

Reviewer Figure 10: Quality control measures for our CTCF-ChIA-PET data in comparison with the ENCODE RAD21 ChIA-PET data. RSC (A) and NSC (B) represent the relative and normalized strand cross correlation coefficient, respectively (Landt et al. 2012). The ENCODE standard for ChIA-PET is indicated as a vertical dashed line. All our replicates are well above the ENCODE standard and within the distribution of the ENCODE ChIA-PET for RAD21.

Reviewer Figure 10

R3.14: The authors try to make the point that CTCF-mediated loops act as independent regulatory units. As shown in figure 2D, the authors chose to check correlation of signal strength by taking k27ac peaks at the CTCF anchor points. Why specifically these regions? What evidence is there that CTCF anchor points are more coordinated than internal enhancer/promoter elements for a given loop?

Authors response: The reason we chose the boundary regions was because we expect a declining correlation with distance between any two peaks. Therefore, by choosing the most distal peaks within a CRD (i.e. the peaks overlapping with each boundary), we are taking the most conservative approach for defining highly correlated groups of peaks.

Authors action: We have added a statement about the motivation of how we defined the CRDs to the main text: *“To identify the highly coordinated chromatin loops we used the correlation of H3K27ac signal of the peaks at both boundaries of a ChIA-PET loop as proxy for the loop’s coordination strength, assuming that if the most distally located peaks within a ChIA-PET loop are correlated all the peaks in-between will be correlated as well (Figure 2C).”*

and to the methods section: *“We are using cohesin ChIA-PET to restrict the potential set of enhancers for each gene by the requirement that they have to share a chromatin regulatory domain. This is based on the observation presented in Figure 2A where we found that H3K27ac peaks within cohesin loops were often highly co-regulated.”*

R3.14: Although the CRD genes identified by the authors represent enrichment in the expected GO categories, caution should be taken in assuming that all enhancer-promoter assignments are bona fide interactions. An experimental validation or additional computational analyses should be provided to strengthen this point. Furthermore, all analyses for CRDs and non-CRDs should be split into actively PAEC-expressed genes and non-expressed genes to avoid biases in the analysis.

Author’s response: We agree with the reviewer that these are not necessarily bona fide interactions, which is also not what we are assuming. All we want to show with this analysis is that by taking genes within CRDs that contain differentially H3K27 acetylated regions we recover more biological signal than when only taking promoters bound by differential H3K27ac into account. This indicates that enhancers are important for recovering the biological signal. In fact, for making the links we are taking the correlations between the H3K27ac signal and gene expression into account and show that these correlations are more positive than randomly chosen genes-H3K27ac peak pairs (see **Figure 2E**).

As for the comment about expressed genes, we always take only the genes into account that were expressed in our PAEC cells (i.e. at least two samples had more than one read for a gene). Similarly, for any GO analysis we always take the set of expressed genes as background.

R3.15: Figure 1F would benefit from removal of overlapping categories (for example, positive regulation of endothelial cell proliferation, endothelial cell proliferation and regulation of endothelial cell proliferation must all be represented by roughly the same set of genes). As shown, this representation gives a false impression of many different processes being significantly affected.

Authors response: (assuming the reviewer refers to Figure 2F) While we agree that there are some highly related terms displayed it becomes very arbitrary when manually filtering out GO terms that seem related. We therefore prefer to keep the full table and make a comment in the figure legend that some of these terms might be highly overlapping in terms of genes.

Authors action: We added the following statement to the legend of Figure 2F: *“Some of the GO terms have very similar sets of genes and are thus not completely independent.”*

Figure 3

R3.16: The authors then applied diffTF to identify transcription factors that show differential activity across the genome in PAH. Although potentially very useful, this method has not, as of now, been peer reviewed and is only available as preprint in bioRxiv. Therefore, the authors conclusions on this section must be taken with caution.

Authors response: The diffTF manuscript is now accepted at Cell Reports. We have attached the accepted manuscript to this submission.

R3.17: diffTF analysis identified differential TF binding, which does not necessarily mean that these TFs are differentially active per se. What is the proposed mechanism for such differential activity if their expression does not change (since the authors state that there are few differentially expressed genes)?

Authors response: Indeed, the reason for which TFs are differentially active while not being significantly differentially expressed can be manifold. It could be due to post transcriptional regulation (such as phosphorylation, localization of the protein, stability of the RNA), however we believe it is more likely that these TFs are slightly differentially expressed, yet to an extent that is not detectable at genome-wide significance. In fact, when we group transcription factors into activators and repressors (using *diffTF* (Berest et al. 2018)) we find that change in activity is reflected by their change in RNA expression i.e. positively correlated for activating TFs and negatively correlated for repressive TFs (see **Reviewer Figure 2**). This indicates that there are small changes on the RNA level of TFs that, when aggregating the signal across all their binding sites, does result in a detectable effect on the chromatin level.

Authors action: We have added part of **Reviewer Fig. 2** as Suppl. Figure 3C and added a statement about the differential TF activity correlating with TF differential expression despite the lack of statistical significance in the latter: *“Reassuringly, when we classified the TFs into activators and repressors (using diffTF in classification mode (Berest et al 2018)) we found an overall significant correlation between differential TF activity and differential expression between patients and controls (SFigure 3B), despite the lack of significantly differentially expressed genes as reported above. This may indicate that small changes in differential expression of a TF may lead to detectable differential activity as measured by differential H3K27ac signal at the TFs target sites.”*

R3.18: Importantly, it is not clear from the manuscript how TF binding sites are identified: is it simply based on TF binding motif mapping, or is there actual ChIP-seq data included? Given the large degree of false-positive rate in motif analysis, this point should be clearer.

Authors response: The TF binding sites were identified based on motifs. We have shown in our diffTF manuscript that this is highly consistent with using only ChIP-Seq validated binding sites. The reason why we have not to be concerned about the huge false positive rate for motif discoveries is that we are simply assuming that the sites that have the motif contain more signal than the sites that do not contain the motif. With this, we are “only” losing signal when considering predicted sites. For more details, we refer the reviewer to the manuscript of diffTF (Berest et al. 2018) (attached is the finally accepted manuscript).

R3.19: Linking back to comments made on previous sections, which mechanism do the authors propose for the detected differential TF activity, when there are no detected changes in the expression of the same TFs (as stated by the authors regarding lack of overall changes in gene expression in PAH samples)?

Authors response: We do think see a small change in gene expression of the TFs, just that it is not detected as statistically significantly different in a differential expression analysis. For more details, we refer the reviewer to our answer to an earlier comment R3.17.

R3.20: This highly speculative section would benefit from some experimental validation to demonstrate that indeed at least some of the predicted TFs with differential activity are binding with different affinity in healthy vs PAH.

Author's response: We agree with the reviewer that experimental validations are key in this study. This is why we have performed the qPCR experiments after stimulating the cells as described in Figure 6. We disagree that validating differential activity could be shown by differential affinity in healthy vs PAH since we do not assume the affinity of the individual binding sites change. What we are proposing is that the overall activity on chromatin of a set of TFs changes. This can have multiple reasons: the TF could be more abundant, it could be more active (e.g. phosphorylated) it could be more nuclear vs cytoplasmic etc. Also, since the changes we see are very complex and seem to affect an entire network of transcriptional regulation, it is unclear whether any small perturbation in the network would have a measurable response. Therefore, we decided to use an independent line of evidence for the TFs (in addition to the literature support that the TFs we identified are highly enriched in known PAH TFs) and showed that the differential activity of TFs is reflected by their change in expression between patients and controls (**Reviewer Figure 8** and response to R3.17).

We further show for YAP1, one of the TFs that our diffTF results predict to be less active in PAH, is indeed less abundant in its active (i.e. phosphorylated) form in PAH when compared to controls (**Reviewer Fig. 4**)

Authors action: We have updated Suppl. Figure 3 as described in response to R3.17. We have also added Reviewer Fig. 4 as **Suppl. Fig. 6B**.

R3.21: Similarly with GO analyses shown earlier, TF motifs should be grouped by TF family, as many TFs of the same family have nearly identical binding sequence preferences (such as ATF1/7 or RFX3/4 to name a couple).

Author's response: We agree with the reviewer that some of the motifs could be explained by related TFs. Instead of clustering them on the figure (which would mess up the signals since some of the similar motifs do show different signals), we have decided to group the TFs by family (which was already done in the previous version of the manuscript). We have now added one more statement to ensure this is appreciated by the readers.

Authors action: We have modified the following sentence in the main text: These newly identified TFs include the AP1 complex (BATF, FOSL1, JUN, JUND, FOSL2 as well as ATF1 and ATF7, all of which have similar binding sites and should thus be considered indistinguishable), members of the RFX family (RFX3, RFX4, again similar motifs), CREB5, and E4F1 with higher activity in patients, and ARID3A, members of the FOX family (FOXL1, FOXG1, FOXJ3, again similar motifs), TBX3 and PITX2 with higher activity in the healthy individuals..

Figure 4

R3.22: This is a very interesting approach to link TFs to their underlying gene networks. However, it seems that the authors are only interested in positive associations between enhancer activity and TF binding. This neglects that a good proportion of TFs can act as repressors of enhancer elements, in which scenario more binding leads to less enhancer activity and target gene expression.

Authors response: We apologize for the confusion. We are indeed interested in repressive interactions and our network takes interactions between TFs and target peaks into account also for negatively correlated TF-H3K27ac interactions. Yet, we do require the H3K27ac mark to positively correlate with the expression of the target gene. This is based on the assumption that activity of enhancers is positively correlated with the expression of the target genes. So what we are missing in our analysis are H3K27ac peaks that negatively correlate with gene expression of a potential target gene. Indeed in Figure 2E we show that negative correlations between H3K27ac peaks and gene expression show a uniform p-value distribution indicating that they are randomly distributed and thus that there is no signal for negative interactions.

Authors action: We have modified the description of the network reconstruction: *“To do so, we devised a correlation based approach linking TFs to their target regulatory element if the regulatory element had a putative binding site of the TF and was significantly (positively or negatively, reflecting activators and repressors) correlated with the expression level of the TF (Figure 4A; Methods).”*

Figure 5

R3.23: Similarly to previous sections, the findings may be potentially very interesting but should be corroborated experimentally.

Authors' response: We agree with the reviewer that some of the findings in Figure 5 should be corroborated experimentally, this is why we have performed the stimulations of the cells followed by qPCR of a couple genes that we predicted from our network (see response to **R3.24**).

Figure 6

R3.24: At the end of the study, the authors tried to experimentally validate some of the findings, aiming to demonstrate that PAECs from PAH patients have a different enhancer landscape, which primes a set of genes to respond aberrantly to stimulation.

This set of experiments is key to the authors point, but should be accompanied by ChIP analysis: are the enhancers under question in this figure over-activated in stimulated conditions, or is the so-called priming of enhancers in PAH all that is required for the observed changes in gene expression?

YAP: this particular TF is precisely regulated by phosphorylation. WB analysis would therefore more informative of the activity of YAP showing levels of phospho-YAP.

Author's response: We agree with the reviewer that it is an interesting question to ask whether the enhancers are active in the steady-state and then even more active upon stimulation, or whether the activity in steady-state is essentially enough to make them activate their target genes. We found that most of the enhancers we tested do indeed get even more active upon stimulation (**Reviewer Fig. 4**)

Similarly, we agree that a WB analysis of the phosphorylated form of YAP is an interesting experiment to do and found that it is indeed much more phosphorylated in the controls than in the patients (**Reviewer Fig. 5**).

Authors action: We have performed ChIP-qPCR to assess the effect on the H3K27ac peaks that are shown in Figure 6 and added **Reviewer Figure 4** as a panel to Figure 6. We have further performed WB analysis of the phosphorylated form of YAP (**Reviewer Figure 5**) and added this to SFigure 6. We have added the following statements to the main text: *“For the TGF-beta stimulation we further showed that the enhancers for the differentially regulated genes become even more differentially active upon stimulation, thus indicating that they are primed in steady-state and gain even more activity upon stimulation (Figure 6B).”* and *“[...] and for YAP1, which is known to be regulated by phosphorylation, we also verified that the phosphorylated form changed in accordance with its RNA level (SFigure 6AB)”*

Minor comments:

Authors response: Thank you for pointing these out, we have corrected them.

- Line 20: “(...) devised a disease-specific enhancer-gene regulatory network based that links transcription factors through enhancers to their target genes.” – this would better as “(...) devised a disease-specific enhancer-gene regulatory network that links transcription factors through enhancers to their target genes.”

Corrected as suggested

- Supplementary Fig1G – figure legend describes a table, this should be revised.
Corrected: *“~~Barchart Table~~ showing the ~~of~~-enrichments of chromatin states in HUVEC for H3K327ac regions detected in PAECs (1st and 2nd column) and differentially modified regions (3rd and 4th column). The plot ~~table~~-provides the adjusted p-value and odd ratios resulting from Fisher's exact test.”*
- Randomised controls should be better explained: for example, which regions were taken, across which genomic interval, and how many times was the shuffling performed.

Corrected: *“[...] again we found more signal for positive correlations, while the p-value distribution for negative correlations and randomized gene-peak links ~~correlations~~ (one round of randomizing the links between genes and peaks while leaving the overall network structure intact)[...]”*

- Axis in graphics should be more self-explanatory, avoiding for example labels such as “Percentage”.

Corrected

- In a few instances H3K27ac is spelled as H3K37 or other.

Corrected

Bibliography

Delaneau, O., Zazhytska, M., Borel, C., et al. 2019. Chromatin three-dimensional interactions mediate genetic effects on gene expression. *Science* 364(6439).

Reviewers' Comments:

Reviewer #1:

Remarks to the Author:

The revised version of the manuscript is a significant improvement compared to the originally submitted version. I thank the authors for their clear and very well organised response. Most of my concerns have been addressed to satisfaction. I only have a few comments remaining.

R3.1 Introduction - The authors are adding an important reference to the most recent large European genetic study in PAH. However, the citation used is incorrect and should be corrected to Graf et al. 2018 (PMID: 29650961). Also, it looks like the references have not been updated according to the changes.

R1.2 Gene panel for screening - The current list of PAH genes includes more than just the six genes that were screened (BMP2, ACVRL1, ENG, CAV1, SMAD1, SMAD4, SMAD9, KCNK3, EIF2AK4, TBX4, ATP13A3, AQP1, SOX17, GDF2, BMP10, KLK1, GCCX). Amongst the most commonly affected genes are BMP2, EIF2AK4 (biallelic) and TBX4, of which two haven't been screened. Unless the authors are able to also look for mutations in the additional genes, they would need to add a statement that additional mutations in other established disease genes that were not screened can't be ruled out. In addition, the authors should state how the screening was performed.

Throughout the manuscript gene symbols need to be formatted italic.

The abbreviation for "small pulmonary artery endothelial cells" is not used uniformly throughout the manuscript, i.e. PAEC versus SPAEC.

Page 16: "Strikingly"

Page 17: "... not only by their impact on or cell surface ..."? Something is missing here.

Page 18: "... This showed two that (a) ..." Swap "two" for "too"

Page 19: "... This particularly interesting" Insert "is" after "This".

Reviewer #2:

Remarks to the Author:

The authors have addressed all my concerns.

Reviewer #4:

Remarks to the Author:

I praise the authors for the efforts to improve their manuscript, which has improved considerably during the revision process. Nevertheless, I still have a few concerns about the claims that the authors make and clarity of data presentation in the manuscript.

Specific points on the manuscript:

1. Despite the efforts be more explicit on the genetic basis of PAH, the authors still failed to cite key literature from the field. In particular, the authors are not citing Rhodes et al. 2019, where genotypes from more than 10,000 individuals have been used to define the genetic determinants of PAH, having identified a number of newly implicated loci.

2. Some claims in the abstract and introduction seem to misleadingly inform that PAH is a special type of multifactorial/polygenic disease. The fact that common polymorphisms that associate with PAH risk are not fully penetrant nor present in all affected individuals is what is observed in every

single polymorphic trait or disease. There seems to be some confusion between what a monogenic form of the disease is, where causal mutations tend to be highly penetrant, and a polygenic form, where disease risk arises from a combination of interactions between multiple genetic variants and environmental risk factors.

This brings us to the pertinent question of whether the authors have tried to bridge the gap between genetics and epigenomics, but looking at allele-specific k27ac deposition in their cohort. Could the differences in enhancer activity detected by the authors in patient samples actually reflect a different genetic background, with higher combined genetic risk, than a direct effect of environmental cues, as the authors claim?

3. The last sentence of the abstract is misleading, and indicates a level of novelty that is not provided by this work. It is now broadly accepted that epigenomic processes and, in particular, transcriptional enhancers, are involved in disease risk. This is particularly true in polygenic diseases, where the vast majority of common polymorphisms associated with these diseases resides within distal open chromatin regions (see Maurano et al. Science 2012). A number of studies with different diseases have demonstrated this and the same have been claimed over the years by large consortia such as the ENCODE and Roadmap Epigenomics. The authors should therefore revise their manuscript and restrict their claims to PAH.

4. The authors should consider revising the use of the word epigenetic by epigenomic, as this term is more accurate to refer to datasets such as histone modification ChIP-seq. Epigenetics is more frequently used in the context of changes, such as DNA methylation, that are passed on from generation to generation. There is no evidence of that in this study.

5. Page 6: the authors claim that the observation that H3K27ac is enriched at TSS is an expected result. Is this really an expected result, given that this is a core mark of active transcriptional enhancers? The authors should provide more information on the datasets to demonstrate the validity of the data and the results of the study. For example, how many k27ac regions overlapped TSS and how many were distally located?

In the same section, the authors say that K4me3 and K27ac were strongly correlated with RNA levels, referring to R² of 0.51 and 0.59. These figures should not be referred to as strong correlations, considering the scale of 0 to 1 in which the R² values are.

The authors also compare the correlations of regulatory elements proximal to TSS versus elements distally located, observing that proximal regions were more correlated with expression. This is expected, but even so, how did the authors define the target genes of k4me1 regions, given that they may represent enhancers, and therefore not necessarily target/regulate their closest genes?

6. Figure S1F: the authors are trying to make the point here that similar both types of PAH show a similar degree of different enhancer landscape in respect to controls. This point would be more easily grasped by reader by using unsupervised clustering analysis. This analysis would also provide a better appreciation of the data.

Minor points on the manuscript:

1. Figure S1E: in Epigenetics, DMR is usually used to refer to differentially methylated region, in the context of DNA methylation studies. The authors should consider revising the use of this acronym in their manuscript.
2. Figure S1F is flipped.
3. Figure S1G legend: there seems to be some confusion in the description of the labelling.

"Distribution of pairwise correlation coefficients for comparing the signal across peaks in pairs of individuals. Correlation coefficients are highest for pairs from the control group, indicating that there is little variation within controls (red). Correlations in pairs of patients (blue) is significantly more variable, indicating increased heterogeneity between patients. Correlation is lowest for patient-control pairs (grey)." According to the inset legend in the figure, red = PAH and blue = control. Patient-control = green?

4. Page 9: Figure 2HF should read Figure 2F

Technical aspects raised during peer review and addressed by the authors in the rebuttal:

R3.4: This study relies of PAECs retrieved from PAH and healthy donors. Histological studies of PAH lung at end stage shows marked regional variation. Regional heterogeneity coupled with 3 to 5 passages of cell culture will undoubtedly add variability to the results and the extent to which the observations are reproducible in other hands is an important consideration.

Authors' response: While we agree that regional heterogeneity and passage may account for some variability, given the similar abnormalities in functional studies with these cells (Sa et al. 2017) and the experimental validations described in Figure 6, we expect that the observations are reproducible.

Unfortunately, this response is not sufficient to demonstrate validity of the samples used in the study and, consequently, of the results obtained. It is well acknowledged in the field that endothelial cells, in general, undergo dedifferentiation after short periods in culture (Milici et al Thum et al. 2000, Lacorre et al. 2004). Which evidence do the authors have that the analysed samples are not showing effects of de-differentiation? At least information on which cell passage was taken for each sample should be provided.

The study that the authors cite here, Sa et al. 2017, was carried out to compare a iPSC model with PAEC, how is this relevant here?

This issue would be better addressed by providing a stratified comparative analysis of RNA and chromatin modification datasets between samples obtained at different passages.

The paper cited here by the authors draws attention to the strong degree of similarity between PAH patient iPSC-derived endothelial cells and PAH PAECs, suggesting that genetic factors are strongly deterministic of PAH, rather than environmental factors, as suggest in this manuscript.

R3.7: To strength the validity of the results, the authors should provide a correlation analysis for signal across all samples/conditions. This analysis will give the reader an appreciation of the dispersion of the signal across samples. The PCAs provided are not enough to make that point. A correlation analysis will also be more relevant than Fig S1B, which could be moved to a supplementary table.

Author's response: We agree with the reviewer that this is an insightful analysis to add. When comparing the global correlations between all individuals we observed similar trends as with the PCA: for the H3K27ac marks the patients were stronger correlated among themselves and distinct from the controls, while no clear difference was visible for the other marks and RNA (see Reviewer Fig. 8).

Authors action: We have added Reviewer Figure 8 as panel in Suppl. Figure 1G and added a statement about it in the main text: "Reassuringly, when we compared the pairwise correlations of the histone or RNA signal between any two individuals we found stronger correlations between pairs of controls than between pairs of patients for the enhancer mark (H3K27ac), the promoter mark (H3K4me3) and RNA (SFigure 1G). This indicates that the lack of significantly differential signal in the promoter and gene expression might be due to the variability of these molecular markers within the group of patients."

I thank the authors for doing this analysis. However, the interpretation is flawed. For RNA, most

PAH samples seem to be more correlated than controls. The dispersion of the RNA data does suggest diminished power to detect statistically significant differences, but it not a specific issue of the PAH samples.

R3.8: The finding of such pervasive chromatin-level alterations without concomitant changes in gene expression is very puzzling, especially after the authors started this section by stating that they observed a strong correlation between RNA levels and K27ac levels. Ideally, the authors should provide two additional pieces of evidence to support this point: 1) screenshot views of the enrichment signal at the genes analysed by qPCR in Fig1F; 2) in a supplementary figure provide a view of enrichment signals across different replicates.

Authors response: We agree that at first it was also very surprising for us to see such pervasive chromatin changes without any significant changes in gene expression. As mentioned earlier though, the fact that we don't detect significant changes in expression doesn't mean there are no changes (see response to R3.1). In fact, in Reviewer Fig. 3B we show that there is a general agreement of the differential expression of TFs and their differential TF activities between patients and controls.

I don't believe that response to R3.1 or Reviewer Fig. 3B address this issue.

With regards to the suggestions of showing the signal of the enhancers for the genes assayed in Fig. 1F, we agree, this is a great suggestion and we have added an extra panel to Figure 1 (Reviewer Figure 9). This also addresses the second suggestion because our "replicates" are in fact the individual patients/controls.

Authors action: We have added Reviewer Fig 9 to Figure 1G to shows the signal of the H3K27ac peaks at the genes that were assayed by qPCR. "G: H3K27ac signal at enhancers of the genes assayed by qPCR in F."

In relation to this, in the manuscript the authors state "we performed validation qPCR in a subset of the individuals (2 IPAH, 2 HPAH, 4 controls) on the baseline expression levels for some genes (Figure 1F), for which we saw a large difference in H3K27ac signal in the promoter region (Figure 1G),"

The authors cannot make claims that there is a large difference in H3K27ac signal in the promoter regions tested. For TSC22D1, the authors present an FDR of 0.6, meaning that the chances of controls and patients being different are as much as the likelihood of getting heads when tossing a coin. Only one of the analysed genes shows an FDR lower or equal to 5% (NOS, FDR: 0.05). This figure raises serious questions about the analysis that was performed, and the thresholds applied to detect significant changes between controls and patients, which is the basis of the whole study.

R3.13: The authors make efforts to assign distal regulatory elements to the right target genes in the chromatin of PAECs, using ChIA-PET to map CTCF-CTCF loops. Although valid, the authors should discuss this choice as opposed to alternative methods that would directly link enhancers to promoters, such as Hi-ChIP for H3K27ac. Also it is confusing that the authors state that they performed a ChIA-PET for CTCF in the main text, while describing ChIA-PET for the Cohesin complex in the Methods – these are conceptually very different experiments, which is correct? To give an appreciation of the quality of the data, the authors should show the data for some example loci and show a distribution of CTCF loop size (as in Figure 2A), but in relation to the corresponding TAD. It is difficult to grasp how different these loops are from previously identified TADs.

Authors response: We apologize for the mistake in the description of the ChIA-PET data (which was mistakenly taken as reference from another manuscript we have been working on at the same time). We used CTCF for the IP and have now updated the description.

Regarding the differences between loops and TADs: it is mainly the size that is very different between these structures (albeit the differences may be more philosophical than quantitative).

TADs tend to be much larger (in the megabase range) than loops (in the kilobase range). While it might be interesting to compare loops and TADs we believe that this is outside the scope of our study since we are mainly using the loops for defining gene-enhancer interactions, yet we agree that it will be nice to show a couple of examples for the ChIA-PET data and CRDs and we have added them now to the Figures (Reviewer Figure 3 added to Figure 2 and SFigure 2). The data quality and a more comprehensive analysis of the ChIA-PET loops - including the ones that we are using in this study, will be described in a companion paper of the current round of ENCODE papers and is accepted at Nature. In Reviewer Fig. 10 we show the quality in terms of the RSC and NSC in relation to the ENCODE ChIA-PET that was recently accepted at Nature. Our two replicates fall well above the ENCODE standard and well within the distribution of ChIA-PET data from ENCODE.

Author's action: We have added a quality control plot that shows the quality of our CTCF ChIA-PET data relative to the RAD21 ChIA-PET data that is part of ENCODE and got recently accepted at Nature. Specifically, we added Reviewer Fig. 10 to Suppl. Figure 2B and the examples in Reviewer Figure 1F to Figure 2G and SFigure 2E. We added the following text to the manuscript: "The data quality was above ENCODE standards and in a similar range to ENCODE ChIA-PET (SFigure 2B)" and "Some examples of genes within CRDs are shown in Figure 2G and SFigure 2E."

There must be a mistake, as no interaction panel has been included in Figure 2.

Figure S2E: It would be useful to include a track with TADs, as suggested previously to gain a better appreciation of how different (in size) these regions really are. This figure makes it visible that not all promoters are baited with this assay (logical, as CTCF is not present in all promoters). The authors should add a section in the discussion in relation to the technical limitations of ChIA-PET in respect to less biased approaches.

R3.14: The authors try to make the point that CTCF-mediated loops act as independent regulatory units. As shown in figure 2D, the authors chose to check correlation of signal strength by taking k27ac peaks at the CTCF anchor points. Why specifically these regions? What evidence is there that CTCF anchor points are more coordinated than internal enhancer/promoter elements for a given loop?

Authors response: The reason we chose the boundary regions was because we expect a declining correlation with distance between any two peaks. Therefore, by choosing the most distal peaks within a CRD (i.e. the peaks overlapping with each boundary), we are taking the most conservative approach for defining highly correlated groups of peaks.

Authors action: We have added a statement about the motivation of how we defined the CRDs to the main text: "To identify the highly coordinated chromatin loops we used the correlation of H3K27ac signal of the peaks at both boundaries of a ChIA-PET loop as proxy for the loop's coordination strength, assuming that if the most distally located peaks within a ChIA-PET loop are correlated all the peaks in-between will be correlated as well (Figure 2C)." and to the methods section: "We are using cohesin ChIA-PET to restrict the potential set of enhancers for each gene by the requirement that they have to share a chromatin regulatory domain. This is based on the observation presented in Figure 2A where we found that H3K27ac peaks within cohesin loops were often highly co-regulated."

Given that assays like HiC or ChIA-PET demonstrate that in fact linear distance means little in terms of gene regulation, as cis-regulatory elements can act over varying distances, this premise does not make much sense.

Regulatory units should be investigated/demonstrated by looking at all elements defined within them. Furthermore, just looking at direction of effects is an imperfect metric, as more than the effect, the degree of effect, i.e. the fold change should be concordant between truly functionally linked elements.

R3.15: Figure 1F would benefit from removal of overlapping categories (for example, positive

regulation of endothelial cell proliferation, endothelial cell proliferation and regulation of endothelial cell proliferation must all be represented by roughly the same set of genes). As shown, this representation gives a false impression of many different processes being significantly affected.

Authors response: (assuming the reviewer refers to Figure 2F) While we agree that there are some highly related terms displayed it becomes very arbitrary when manually filtering out GO terms that seem related. We therefore prefer to keep the full table and make a comment in the figure legend that some of these terms might be highly overlapping in terms of genes.

Authors action: We added the following statement to the legend of Figure 2F: "Some of the GO terms have very similar sets of genes and are thus not completely independent."

REVIGO or GO Trimming would take care of this problem.

All other points I raised previously have been fully addressed.

Point-by-point response to reviewers' comments:

Below we have addressed the remaining comments of the reviewers (also marked in red in the revised manuscript). In addition, we have significantly shortened the text of the manuscript to fit with Nature Communications requirements of 5000 words for the main text. These general changes (independent of any reviewer comments) are in word track-change mode.

We have color-coded our response as follows:

- Reviewer comments in black
- Our response to a previous round of review in blue
- Our response to this round of review in red

Reviewer #1 (Remarks to the Author):

The revised version of the manuscript is a significant improvement compared to the originally submitted version. I thank the authors for their clear and very well organised response. Most of my concerns have been addressed to satisfaction. I only have a few comments remaining.

R3.1 Introduction - The authors are adding an important reference to the most recent large European genetic study in PAH. However, the citation used is incorrect and should be corrected to Graf et al. 2018 (PMID: 29650961). Also, it looks like the references have not been updated according to the changes.

Authors response and action: We thank the referee for pointing this out and apologize for the mistake. We have now correctly updated the references.

R1.2 Gene panel for screening - The current list of PAH genes includes more than just the six genes that were screened (BMP2, ACVRL1, ENG, CAV1, SMAD1, SMAD4, SMAD9, KCNK3, EIF2AK4, TBX4, ATP13A3, AQP1, SOX17, GDF2, BMP10, KLK1, GCCX). Amongst the most commonly affected genes are BMP2, EIF2AK4 (biallelic) and TBX4, of which two haven't been screened. Unless the authors are able to also look for mutations in the additional genes, they would need to add a statement that additional mutations in other established disease genes that were not screened can't be ruled out. In addition, the authors should state how the screening was performed.

Authors response: We thank the referee for pointing this out. These additional genes have not been screened in our patients and we have added the remark that patients were not screened for these additional known genes.

Authors action: "All patients were screened for a panel of PAH genes that include BMP2, ACVRL1, ENG, SMAD9, CAV1, KCNK3 and had none of these mutations except for two BMP2 mutation. The BMP2 carriers are indicated in Suppl. Table 1. We acknowledge that by the date of publication additional PAH genes have been added to the panel, for which our patients have not been screened (SMAD1, SMAD4, KCNK3, EIF2AK4, TBX4, ATP13A3, AQP1, SOX17, GDF2, BMP10, KLK1, GCCX)"

Throughout the manuscript gene symbols need to be formatted italic.

Authors response and action: We thank the referee for pointing this out and have corrected all occurrences of gene names.

The abbreviation for "small pulmonary artery endothelial cells" is not used uniformly throughout the manuscript, i.e. PAEC versus SPAEC.

Authors response and action: We thank the referee for pointing this out and have now used the abbreviation (PAEC) consistently.

Page 16: "Strikingly"

Page 17: "... not only by their impact on or cell surface ..."? Something is missing here.

Page 18: "... This showed two that (a) ..." Swap "two" for "too"

Page 19: "... This particularly interesting" Insert "is" after "This".

Authors response and action: We thank the referee for pointing out these typos and have corrected them all.

Reviewer #2 (Remarks to the Author):

The authors have addressed all my concerns.

Authors response: we thank the referee for assessing our manuscript and helping in improving it.

Reviewer #4 (Remarks to the Author):

I praise the authors for the efforts to improve their manuscript, which has improved considerably during the revision process. Nevertheless, I still have a few concerns about the claims that the authors make and clarity of data presentation in the manuscript.

Authors response: we thank the referee for the praise and the thorough assessment of our manuscript.

Specific points on the manuscript:

1. Despite the efforts be more explicit on the genetic basis of PAH, the authors still failed to cite key literature from the field. In particular, the authors are not citing Rhodes et al. 2019, where genotypes from more than 10,000 individuals have been used to define the genetic determinants of PAH, having identified a number of newly implicated loci.

Authors response and action: we thank the referee for pointing this out and have added the reference to the paper: "*SOX17, which was only recently identified as PAH TF (Rhodes et al. 2019; Gräf et al. 2018) was also significantly miss-regulated in our PAH patients (Supplementary Table 5).*" (p10)

2. Some claims in the abstract and introduction seem to misleadingly inform that PAH is a special type of multifactorial/polygenic disease. The fact that common polymorphisms that associate with PAH risk are not fully penetrant nor present in all affected individuals is what is observed in every single polymorphic trait or disease. There seems to be some confusion between what a monogenic form of the disease is, where causal mutations tend to be highly

penetrant, and a polygenic form, where disease risk arises from a combination of interactions between multiple genetic variants and environmental risk factors.

Authors response: we thank the reviewer for pointing out this confusion.

Authors action: We have changed the following sentence in the introduction: "~~Thus However, as in polygenic diseases in general, in the vast majority of IPAH patients carry no known mutation has been described and 80% of family members with a *BMP2* mutation do not have HPAH.~~" (p 4)

This brings us to the pertinent question of whether the authors have tried to bridge the gap between genetics and epigenomics, but looking at allele-specific k27ac deposition in their cohort. Could the differences in enhancer activity detected by the authors in patient samples actually reflect a different genetic background, with higher combined genetic risk, than a direct effect of environmental cues, as the authors claim?

Authors response: we thank the reviewer for this suggestion. Unfortunately, we do not have the genotypes of these individuals available and an allele-specific analysis is therefore not possible. Also, the cohort may be too small to do such genetic analysis.

3. The last sentence of the abstract is misleading, and indicates a level of novelty that is not provided by this work. It is now broadly accepted that epigenomic processes and, in particular, transcriptional enhancers, are involved in disease risk. This is particularly true in polygenic diseases, where the vast majority of common polymorphisms associated with these diseases resides within distal open chromatin regions (see Maurano et al. Science 2012). A number of studies with different diseases have demonstrated this and the same have been claimed over the years by large consortia such as the ENCODE and Roadmap Epigenomics. The authors should therefore revise their manuscript and restrict their claims to PAH.

Authors response: we agree with the reviewer that the notion of epigenetics and gene regulatory processes being involved in diseases is not a novel concept. In our opinion, the sentence the referee is referring to has absolutely no claim of novelty. It is one of the important take-home messages that we want to convey with this study.

In our opinion there is still a large disconnect between claiming that gene regulation has to be involved because GWAS SNPs lie in enhancers and regulatory elements derived mainly from healthy samples or cell lines, and actually showing that clinical samples differ in their epigenetic profile – ours being one of the few that do show epigenetics profiles for patients.

Overall, there is a lot of focus on analysing changes in gene expression levels and our study shows that it is important not only to consider gene expression differences but also taking into account the differences in the epigenetic and gene regulatory landscape in general. We agree, that we have only looked at PAH as a special case, and we therefore have changed the sentence in question to be more specific to PAH.

Authors action: We have changed the last sentence of the abstract to: "*Our study highlights the role of chromatin state-epigenetics and enhancers in polygenic, multifactorial diseases that can be polygenic or multifactorial, such as PAH, and suggests that therapeutic*

approaches should be assessed by their ability to reverse such disease-specific epigenetic alterations.” (p1)

4. The authors should consider revising the use of the word epigenetic by epigenomic, as this term is more accurate to refer to datasets such as histone modification ChIP-seq. Epigenetics is more frequently used in the context of changes, such as DNA methylation, that are passed on from generation to generation. There is no evidence of that in this study.

Authors response: we agree with the reviewer that epigenetics is often used in the context of DNA methylation, and we have tried to be more specific whenever possible, mentioning directly the histone marks instead of abbreviating as epigenetics.

However, we disagree with the reviewer for the term epigenomics vs epigenetics. The definition of epigenetics has nothing to do with DNA methylation, but rather with whether there is a cellular memory of some sort that can be transmitted through to the next generation of cells. In a recent “User guide to the ambiguous word ‘epigenetics’” (<https://www.nature.com/articles/nrm.2017.135>) it says: one should use epigenetics to describe “a cellular memory, persistent homeostasis in the absence of the original perturbation or an effect on cell fate not attributable to changes in DNA sequence“, which is what we are describing.

Author’s action: we have removed the term epigenetics throughout the manuscript in the all instances where we thought it is not directly evident that we are talking about histone marks:

- Abstract: *“Our study highlights the role of chromatin state-epigenetics and enhancers in polygenic, multifactorial diseases ~~that can be polygenic or multifactorial~~, such as PAH, and suggests that therapeutic approaches should be assessed by their ability to reverse such disease-specific epigenetic alterations.” (p1)*
- Introduction to our experiments: *“[...] performed gene expression (RNA-Seq), chromatin marks ChIP-Seq epigenetic-(H3K27ac, H3K4me1, H3K4me3ChIP-Seq), and chromatin interaction (ChIA-PET) profiling in the affected pulmonary arterial endothelial cells (PAECs)[...]” (p3)*
- Results: *“This coordination of PAH-associated chromatin epigenetic changes across entire ~~chromatin~~ CTCF loops suggested[...]*” (p7)
- Results: *“All together our data indicate that the active chromatin epigenetic landscape in PAH patients is specifically altered at enhancers [...]” (p14)*

5. Page 6: the authors claim that the observation that H3K27ac is enriched at TSS is an expected result. Is this really an expected result, given that this is a core mark of active transcriptional enhancers? The authors should provide more information on the datasets to demonstrate the validity of the data and the results of the study. For example, how many k27ac regions overlapped TSS and how many were distally located?

Authors response and action: the H3K27ac mark is a well-known mark of active promoters and enhancers. That is why we mention that it is an expected enrichment at transcription start sites. See Figure 1 in the paper by Ernst and Kellis that describe the widely used and ENCODE approved algorithm chromHMM (<https://www.nature.com/articles/nprot.2017.124>),

which provides chromatin state annotation based on combinations of histone marks. We find the question of how many K27ac regions overlap with a TSS irrelevant for the current study since we clearly show the expected enrichment and K27ac in promoters is not the focus of the study.

In the same section, the authors say that K4me3 and K27ac were strongly correlated with RNA levels, referring to R2 of 0.51 and 0.59. These figures should not be referred to as strong correlations, considering the scale of 0 to 1 in which the R2 values are.

Authors response and action: we agree with the referee that the interpretation of correlation values is always subjective. We have therefore removed the word “strongly” from the sentence. We want to point out though that we are reporting R (not R2 as assumed by the referee), for which the scale goes from -1 to 1.

The authors also compare the correlations of regulatory elements proximal to TSS versus elements distally located, observing that proximal regions were more correlated with expression. This is expected, but even so, how did the authors define the target genes of k4me1 regions, given that they may represent enhancers, and therefore not necessarily target/regulate their closest genes?

Authors response and action: (we assume the reviewer refers to Reviewer Figure 3D for this comment). We apologize for the unclear labelling of the y-axis, we here only used H3K27ac signal for the correlations with RNA expression. It is an exhaustive analysis comparing all pairwise positive correlations of peaks and genes on the same chromosome. We apologize for not having labelled the y-axis properly: it is the distance in log2 bp (see below)

6. Figure S1F: the authors are trying to make the point here that similar both types of PAH show a similar degree of different enhancer landscape in respect to controls. This point would be more easily grasped by reader by using unsupervised clustering analysis. This analysis would also provide a better appreciation of the data.

Authors response and action: we agree with the referee that unsupervised clustering would

be a valid alternative to the correlation analysis between the fold-change levels. However, given that the point we are trying to make is very minor – i.e. that we are not able to see any specific difference between hPAH and iPAH, we believe that the current analysis is much more direct at showing that there is no interpretable difference between iPAH and hPAH. We also mention in the manuscript that we are unable to say whether this is because there is indeed no difference, or because the sample number of hPAH is too small. The reason we are not adding more analysis to this, is because we are not confident to draw any conclusion from two samples with the hPAH.

Minor points on the manuscript:

1. Figure S1E: in Epigenetics, DMR is usually used to refer to differentially methylated region, in the context of DNA methylation studies. The authors should consider revising the use of this acronym in their manuscript.

Authors response and action: we thank the referee for pointing this out (assuming the reviewer is pointing at Figure S2A). We have changed this abbreviation to “differentially H3K27ac peak” and “differentially H3K27acetylated peaks” in the panel and figure legend of Figure S2A respectively.

2. Figure S1F is flipped.

Authors response and action: thank you for pointing this out - we have corrected it.

3. Figure S1G legend: there seems to be some confusion in the description of the labelling. “Distribution of pairwise correlation coefficients for comparing the signal across peaks in pairs of individuals. Correlation coefficients are highest for pairs from the control group, indicating that there is little variation within controls (red). Correlations in pairs of patients (blue) is significantly more variable, indicating increased heterogeneity between patients. Correlation is lowest for patient-control pairs (grey).” According to the inset legend in the figure, red = PAH and blue = control. Patient-control = green?

Authors response and action: thank you for pointing this out - we have corrected the Figure.

4. Page 9: Figure 2HF should read Figure 2F

Authors response and action: thank you for pointing this out - we have corrected this.

Technical aspects raised during peer review and addressed by the authors in the rebuttal:

R3.4: This study relies of PAECs retrieved from PAH and healthy donors. Histological studies of PAH lung at end stage shows marked regional variation. Regional heterogeneity coupled with 3 to 5 passages of cell culture will undoubtedly add variability to the results and the extent to which the observations are reproducible in other hands is an important consideration.

Authors' response: While we agree that regional heterogeneity and passage may account for some variability, given the similar abnormalities in functional studies with these cells (Sa et al. 2017) and the experimental validations described in Figure 6, we expect that the observations are reproducible.

Unfortunately, this response is not sufficient to demonstrate validity of the samples used in the study and, consequently, of the results obtained. It is well acknowledged in the field that endothelial cells, in general, undergo dedifferentiation after short periods in culture (Milici et al Thum et al. 2000, Lacorre et al. 2004). Which evidence do the authors have that the analysed samples are not showing effects of de-differentiation? At least information on which cell passage was taken for each sample should be provided.

The study that the authors cite here, Sa et al. 2017, was carried out to compare a iPSC model with PAEC, how is this relevant here?

This issue would be better addressed by providing a stratified comparative analysis of RNA and chromatin modification datasets between samples obtained at different passages. The paper cited here by the authors draws attention to the strong degree of similarity between PAH patient iPSC-derived endothelial cells and PAH PAECs, suggesting that genetic factors are strongly deterministic of PAH, rather than environmental factors, as suggest in this manuscript.

Authors response: The reviewer raises a good point to try to better understand why previous studies (Rhodes and Sa manuscripts) established gene expression changes between PAH patient and control cells, not seen in this cohort where epigenetic changes are abundant. While the reviewer suggests that the epigenetic changes observed in this study, and the lack of gene expression changes compared to previous studies may be the result of de-differentiation of the endothelial cells, there is no evidence of this. We restricted our endothelial cell cultures to passages 3-5 in this manuscript as well as in in the other two manuscripts where we did show changes in gene expression between controls and PAH patient cells. We have sometimes observed evidence of de-differentiation in endothelial cell cultures, but almost exclusively at or beyond passage 8. We also confirmed no changes in the morphology of the cells that would indicate de-differentiation, i.e., loss of the cobblestone monolayer or loss of sharp VE-cadherin cell boundaries, no loss of CD31 or CD144 or loss of acetylated LDL uptake.

Author's action: We have revised the text by adding a few sentences to the Methods on page 19: *"As in our previous studies (Rhodes et al and Sa et al), we restricted the endothelial cells used in these experiments to passages 3-5 as we have found no evidence of de-differentiation in these early passages. Lack of de-differentiation was judged by a tight cobblestone morphology, prominent CD144 (ve-cadherin) cell boundaries between all cells and uniform uptake of acetylated LDL."* (p19)

R3.7: To strength the validity of the results, the authors should provide a correlation analysis for signal across all samples/conditions. This analysis will give the reader an appreciation of the dispersion of the signal across samples. The PCAs provided are not enough to make that point. A correlation analysis will also be more relevant than Fig S1B, which could be moved to a supplementary table.

Author's response: We agree with the reviewer that this is an insightful analysis to add. When comparing the global correlations between all individuals we observed similar trends as with the PCA: for the H3K27ac marks the patients were stronger correlated among themselves and distinct from the controls, while no clear difference was visible for the other marks and RNA (see Reviewer Fig. 8).

Authors action: We have added Reviewer Figure 8 as panel in Suppl. Figure 1G and added

a statement about it in the main text: "Reassuringly, when we compared the pairwise correlations of the histone or RNA signal between any two individuals we found stronger correlations between pairs of controls than between pairs of patients for the enhancer mark (H3K27ac), the promoter mark (H3K4me3) and RNA (SFigure 1G). This indicates that the lack of significantly differential signal in the promoter and gene expression might be due to the variability of these molecular markers within the group of patients."

I thank the authors for doing this analysis. However, the interpretation is flawed. For RNA, most PAH samples seem to be more correlated than controls. The dispersion of the RNA data does suggest diminished power to detect statistically significant differences, but it not a specific issue of the PAH samples.

Authors response: We thank the reviewer for the thorough reading of the figure. This is indeed our mistake in the interpretation and we have now changed the interpretation to what the reviewer was suggesting, that indeed that the dispersion for RNA is much greater than for H3K27ac and that this may be one reason of why we don't find any differences on the expression level.

Authors action: We have changed the sentence describing this on p7 from: ~~Reassuringly, when we compared the pairwise correlations of the histone or RNA signal between any two individuals, we found stronger correlations between pairs of controls than between pairs of patients for the enhancer mark (H3K27ac), the promoter mark (H3K4me3) and RNA (SFigure 1G). This indicates that the lack of significantly differential signal in the promoter and gene expression might be due to the variability of these molecular markers within the group of patients.~~

to: "When we compared pair-wise correlations between individuals for histone marks and RNA we found a significantly higher variation for RNA pairs, regardless of whether the individuals came from PAH or controls, indicating that the lack of differentially expressed genes may partially be explained by sample heterogeneity (Supplementary Figure 1G)." (p6)

We have also rewritten the legend for Supplementary Figure 1G: "G) Distribution of pairwise correlation coefficients for comparing the signal across peaks or genes between pairs of individuals. Correlations are shown for pairs of patients (red), pairs of controls (blue) and pairs between patients and control (grey). RNA is least correlated between any pairs, suggesting heterogeneity as one reason for the lack in differential expression signal."

R3.8: The finding of such pervasive chromatin-level alterations without concomitant changes in gene expression is very puzzling, especially after the authors started this section by stating that they observed a strong correlation between RNA levels and K27ac levels. Ideally, the authors should provide two additional pieces of evidence to support this point: 1) screenshot views of the enrichment signal at the genes analysed by qPCR in Fig1F; 2) in a supplementary figure provide a view of enrichment signals across different replicates.

Authors response: We agree that at first it was also very surprising for us to see such pervasive chromatin changes without any significant changes in gene expression. As mentioned earlier though, the fact that we don't detect significant changes in expression doesn't mean there are no changes (see response to R3.1). In fact, in Reviewer Fig. 3B we show that there is a general agreement of the differential expression of TFs and their differential TF activities between patients and controls.

I don't believe that response to R3.1 or Reviewer Fig. 3B address this issue.

Authors response and action: We apologize for the miss-referencing. We meant to refer to "Summary of additional experiments / analyses" **point 2**" and corresponding Reviewer **Figure 2B** (see below).

Reviewer Figure 2

Summary of additional analysis point 2:

We showed that, despite the lack of significantly differentially expressed genes, the changes in activities for TFs are correlated with their change in expression level, which indicates that there our sample may be to heterogeneous for detecting significant changes in RNA. To do so, we have classified TFs into activators and repressors (using *diffTF* (Berest et al. 2019)) based on whether their expression correlates positively or negatively with the H3K27ac signal at their putative target sites. This revealed that indeed for TFs classified as activators there is a positive correlation between the fold change in RNA and TF activity between PAH and controls (**Reviewer Fig. 2** – added to Suppl. Figure 3).

With regards to the suggestions of showing the signal of the enhancers for the genes assayed in Fig. 1F, we agree, this is a great suggestion and we have added an extra panel to Figure 1 (Reviewer Figure 9). This also addresses the second suggestion because our "replicates" are in fact the individual patients/controls.

Authors action: We have added Reviewer Fig 9 to Figure 1G to shows the signal of the

H3K27ac peaks at the genes that were assayed by qPCR. “G: H3K27ac signal at enhancers of the genes assayed by qPCR in F.”

In relation to this, in the manuscript the authors state “we performed validation qPCR in a subset of the individuals (2 IPAH, 2 HPAH, 4 controls) on the baseline expression levels for some genes (Figure 1F), for which we saw a large difference in H3K27ac signal in the promoter region (Figure 1G),”

The authors cannot make claims that there is a large difference in H3K27ac signal in the promoter regions tested. For TSC22D1, the authors present an FDR of 0.6, meaning that the chances of controls and patients being different are as much as the likelihood of getting heads when tossing a coin. Only one of the analysed genes shows an FDR lower or equal to 5% (NOS, FDR: 0.05). This figure raises serious questions about the analysis that was performed, and the thresholds applied to detect significant changes between controls and patients, which is the basis of the whole study.

Authors response: We agree with the reviewer that our statement regarding Figure 1G is misleading.

Authors action: We have corrected this to say: “we performed validation qPCR in a subset of the individuals (2 IPAH, 2 HPAH, 4 controls) on the baseline expression levels for a some genes (Figure 1F) that were selected to have a range of differential H3K27ac signals in their promoters for which we saw a large difference in H3K27ac signal in the promoter region (Figure 1G)” (p6)

R3.13: The authors make efforts to assign distal regulatory elements to the right target genes in the chromatin of PAECs, using ChIA-PET to map CTCF-CTCF loops. Although valid, the authors should discuss this choice as opposed to alternative methods that would directly link enhancers to promoters, such as Hi-ChIP for H3K27ac. Also it is confusing that the authors state that they performed a ChIA-PET for CTCF in the main text, while describing ChIA-PET for the Cohesin complex in the Methods – these are conceptually very different experiments, which is correct?

To give an appreciation of the quality of the data, the authors should show the data for some example loci and show a distribution of CTCF loop size (as in Figure 2A), but in relation to the corresponding TAD. It is difficult to grasp how different these loops are from previously identified TADs.

Authors response: We apologize for the mistake in the description of the ChIA-PET data (which was mistakenly taken as reference from another manuscript we have been working on at the same time). We used CTCF for the IP and have now updated the description. Regarding the differences between loops and TADs: it is mainly the size that is very different between these structures (albeit the differences may be more philosophical than quantitative). TADs tend to be much larger (in the mega base range) than loops (in the kilobase range). While it might be interesting to compare loops and TADs we believe that this is outside the scope of our study since we are mainly using the loops for defining gene-enhancer interactions, yet we agree that it will be nice to show a couple of examples for the ChIA-PET data and CRDs and we have added them now to the Figures (Reviewer Figure 3 added to Figure 2 and SFigure 2). The data quality and a more comprehensive analysis of the ChIA-PET loops - including the ones that we are using in this study, will be described in a companion paper of the current round of ENCODE papers and is accepted at Nature. In Reviewer Fig. 10 we show the quality in terms of the RSC and NSC in relation to the

ENCODE ChIA-PET that was recently accepted at Nature. Our two replicates fall well above the ENCODE standard and well within the distribution of ChIA-PET data from ENCODE.

Author's action: We have added a quality control plot that shows the quality of our CTCF ChIA-PET data relative to the RAD21 ChIA-PET data that is part of ENCODE and got recently accepted at Nature. Specifically, we added Reviewer Fig. 10 to Suppl. Figure 2B and the examples in Reviewer Figure 1F to Figure 2G and SFigure 2E. We added the following text to the manuscript: "The data quality was above ENCODE standards and in a similar range to ENCODE ChIA-PET (SFigure 2B)" and "Some examples of genes within CRDs are shown in Figure 2G and SFigure 2E."

There must be a mistake, as no interaction panel has been included in Figure 2. Figure S2E: It would be useful to include a track with TADs, as suggested previously to gain a better appreciation of how different (in size) these regions really are. This figure makes it visible that not all promoters are baited with this assay (logical, as CTCF is not present in all promoters). The authors should add a section in the discussion in relation to the technical limitations of ChIA-PET in respect to less biased approaches.

Authors response: We thank the reviewer for pointing this out, we have indeed uploaded the old version of Figure 2. See below for the correct Figure 2, which shows the interactions in panel H. For the ChIA-PET we have added a point in the discussion.

Authors action: We have added a sentence about the pitfalls of ChIA-PET in the discussion on p17: "*We also note that one pitfall of using CTCF-ChIA-PET as a basis for calling chromatin regulatory domains is that we will be blind to any gene loop that is not demarcated by CTCF.*" (p 17)

With regards to the comment about TADs, to our knowledge there is no "TAD-track" available for our specific cell type. Also calling TADs, in the experience of the authors, typically relies on a number of arbitrary thresholds to define TAD boundaries. We therefore find it irrelevant to this specific study to add tracks that are (a) based on arbitrary thresholds and (b) not even from our specific cell type – which makes them very difficult to interpret. Therefore, we disagree with the reviewer that it would be helpful to add such tracks.

Figure 2

R3.14: The authors try to make the point that CTCF-mediated loops act as independent regulatory units. As shown in figure 2D, the authors chose to check correlation of signal strength by taking k27ac peaks at the CTCF anchor points. Why specifically these regions? What evidence is there that CTCF anchor points are more coordinated than internal enhancer/promoter elements for a given loop?

Authors response: The reason we chose the boundary regions was because we expect a declining correlation with distance between any two peaks. Therefore, by choosing the most distal peaks within a CRD (i.e. the peaks overlapping with each boundary), we are taking the most conservative approach for defining highly correlated groups of peaks.

Authors action: We have added a statement about the motivation of how we defined the CRDs to the main text: “To identify the highly coordinated chromatin loops we used the correlation of H3K27ac signal of the peaks at both boundaries of a ChIA-PET loop as proxy for the loop’s coordination strength, assuming that if the most distally located peaks within a ChIA-PET loop are correlated all the peaks in-between will be correlated as well (Figure 2C).”

and to the methods section: “We are using cohesin ChIA-PET to restrict the potential set of enhancers for each gene by the requirement that they have to share a chromatin regulatory domain. This is based on the observation presented in Figure 2A where we found that H3K27ac peaks within cohesin loops were often highly co-regulated.”

Given that assays like HiC or ChIA-PET demonstrate that in fact linear distance means little in terms of gene regulation, as cis-regulatory elements can act over varying distances, this premise does not make much sense.

Authors response and action: We disagree with the referee in this point. ChIA-PET and HiC data indeed show that most of the interactions occur very locally (as evidenced by the strong signal that is present along the diagonal in the typical HiC plots). Below is an illustration of the chromatin regulatory domain (CRD) definition by Delaneau et al 2019 (<https://science.sciencemag.org/content/364/6439/eaat8266.long>), which shows that regulatory domains are locally highly structured and that enhancers are rarely interacting across CRDs (which were here defined as correlation of H3K27ac signal across >300 individuals). The figure also shows that loops (here based on Hi-C) are usually larger than CRDs, thus our definition of correlating boundaries of ChIA-PET loops seem appropriate to us.

Regulatory units should be investigated/demonstrated by looking at all elements defined within them.

Authors response: We thank the reviewer for pointing out that this was not clearly described in the manuscript, since it is exactly what we have done. We have clarified our approach and the assumptions leading to it, in the main text.

Authors action: We have modified and added text in the main manuscript to describe our approach and motivation for doing so more clearly: *“This coordination of PAH-associated chromatin changes across entire CTCF loops suggested that they may act as independent regulatory units. Thus, instead of requiring a direct enhancer-promoter contact, we sought that we can use the highly coordinated loops, that we termed chromatin regulatory domains (CRD), as a basis for searching for enhancer-target gene interactions. To identify the CRD, we used the correlation of H3K27ac signal at both boundaries of a ChIA-PET loop as proxy for the loop’s coordination strength, assuming that if the most distally located H3K27ac peaks of a loop are correlated, all peaks in-between will be correlated as well. A loop was then defined as CRD when the boundary H3K27ac peaks were positively correlated with a nominal*

p-value <0.05 (Figure 2C). A similar concept of CRDs has recently been proposed in lymphoblastoid cell lines [...]"

Furthermore, just looking at direction of effects is an imperfect metric, as more than the effect, the degree of effect, i.e. the fold change should be concordant between truly functionally linked elements.

Authors response: We agree, this method is imperfect – as are most methods. Since we are using the chromatin regulatory domain definition as a filtering criteria for enhancer-promoter interactions the worst that can happen by considering only the anchor points is that we are testing more connections than necessary – thus slightly reducing our power. Given that it is non-trivial to implement the proposition by the referee, and that it would also rely on a similarly arbitrary threshold, we argue that this analysis is out of scope.

R3.15: Figure 1F would benefit from removal of overlapping categories (for example, positive regulation of endothelial cell proliferation, endothelial cell proliferation and regulation of endothelial cell proliferation must all be represented by roughly the same set of genes). As shown, this representation gives a false impression of many different processes being significantly affected.

Authors response: (assuming the reviewer refers to Figure 2F) While we agree that there are some highly related terms displayed it becomes very arbitrary when manually filtering out GO terms that seem related. We therefore prefer to keep the full table and make a comment in the figure legend that some of these terms might be highly overlapping in terms of genes.

Authors action: We added the following statement to the legend of Figure 2F: “Some of the GO terms have very similar sets of genes and are thus not completely independent.”

REVIGO or GO Trimming would take care of this problem.

Authors response: We thank the reviewer for this suggestion, yet we disagree that showing the full list of GO terms is a “problem”. We believe it is clearer to the reader to see the full list of terms instead of selecting specific terms based on algorithms that also rely on some arbitrary criteria that are difficult for the reader to judge.

All other points I raised previously have been fully addressed.

Bibliography

Berest, I., Arnold, C., Reyes-Palomares, A., et al. 2019. Quantification of Differential Transcription Factor Activity and Multiomics-Based Classification into Activators and Repressors: diffTF. *Cell reports* 29(10), p. 3147–3159.e12.

Gräf, S., Haimel, M., Bleda, M., et al. 2018. Identification of rare sequence variation underlying heritable pulmonary arterial hypertension. *Nature Communications* 9(1), p. 1416.

Rhodes, C.J., Batai, K., Bleda, M., et al. 2019. Genetic determinants of risk in pulmonary arterial hypertension: international genome-wide association studies and meta-analysis. *The Lancet. Respiratory medicine* 7(3), pp. 227–238.

Reviewers' Comments:

Reviewer #4:

Remarks to the Author:

The authors have addressed all my concerns.